# A broadly distributed toxin family mediates contact-dependent antagonism between gram-positive bacteria

John C Whitney[1†], S Brook Peterson[1], Jungyun Kim[1], Manuel Pazos[2], Adrian J Verster[3], Matthew C Radey[1], Hemantha D Kulasekara[1], Mary Q Ching[1], Nathan P Bullen[4,5], Diane Bryant[6], Young Ah Goo[7], Michael G Surette[4,5,8], Elhanan Borenstein[3,9,10], Waldemar Vollmer[2], Joseph D Mougous[1,11]*

[1]Department of Microbiology, University of Washington School of Medicine, Seattle, United States; [2]Centre for Bacterial Cell Biology, Institute for Cell and Molecular Biosciences, Newcastle University, Newcastle, United Kingdom; [3]Department of Genome Sciences, University of Washington, Seattle, United States; [4]Michael DeGroote Institute for Infectious Disease Research, McMaster University, Hamilton, Canada; [5]Department of Biochemistry and Biomedical Sciences, McMaster University, Hamilton, Canada; [6]Experimental Systems Group, Advanced Light Source, Berkeley, United States; [7]Northwestern Proteomics Core Facility, Northwestern University, Chicago, United States; [8]Department of Medicine, Farncombe Family Digestive Health Research Institute, McMaster University, Hamilton, Canada; [9]Department of Computer Science and Engineering, University of Washington, Seattle, United States; [10]Santa Fe Institute, Santa Fe, United States; [11]Howard Hughes Medical Institute, University of Washington School of Medicine, Seattle, United States

*For correspondence: mougous@uw.edu

Present address: [†]Department of Biochemistry and Biomedical Sciences, McMaster University, Hamilton, Canada

Competing interests: The authors declare that no competing interests exist.

**Abstract** The Firmicutes are a phylum of bacteria that dominate numerous polymicrobial habitats of importance to human health and industry. Although these communities are often densely colonized, a broadly distributed contact-dependent mechanism of interbacterial antagonism utilized by Firmicutes has not been elucidated. Here we show that proteins belonging to the LXG polymorphic toxin family present in *Streptococcus intermedius* mediate cell contact- and Esx secretion pathway-dependent growth inhibition of diverse Firmicute species. The structure of one such toxin revealed a previously unobserved protein fold that we demonstrate directs the degradation of a uniquely bacterial molecule required for cell wall biosynthesis, lipid II. Consistent with our functional data linking LXG toxins to interbacterial interactions in *S. intermedius*, we show that LXG genes are prevalent in the human gut microbiome, a polymicrobial community dominated by Firmicutes. We speculate that interbacterial antagonism mediated by LXG toxins plays a critical role in shaping Firmicute-rich bacterial communities.

## Introduction

Bacteria in polymicrobial environments must persist in the face of frequent physical encounters with competing organisms. Studies have revealed Gram-negative bacterial species contend with this threat by utilizing pathways that mediate antagonism toward contacting bacterial cells (*Konovalova and Søgaard-Andersen, 2011*). For instance, Proteobacteria widely employ contact-dependent inhibition (CDI) to intoxicate competitor cells that share a high degree of phylogenetic

**eLife digest** Most bacteria live in densely colonized environments, such as the human gut, in which they must constantly compete with other microbes for space and nutrients. As a result, bacteria have evolved a wide array of strategies to directly fight their neighbors. For example, some bacteria release antimicrobial compounds into their surroundings, while others 'inject' protein toxins directly into adjacent cells.

Bacteria can be classified into two groups known as Gram-positive and Gram-negative. Previous studies found that Gram-negative bacteria inject toxins into neighboring cells, but no comparable toxins in Gram-positive bacteria had been identified. Before a bacterium can inject molecules into an adjacent cell, it needs to move the toxins from its interior to the cell surface. It had been suggested that a transport system in Gram-positive bacteria called the Esx pathway may export toxins known as LXG proteins. However, it was not clear whether these proteins help Gram-positive bacteria to compete against other bacteria.

Whitney et al. studied the LXG proteins in Gram-positive bacteria known as Firmicutes. The experiments reveal that Firmicutes found in the human gut possess LXG genes. A Firmicute known as *Streptococcus intermedius* produces three LXG proteins that are all toxic to bacteria. To avoid being harmed by its own LXG proteins, *S. intermedius* also produces matching antidote proteins. Further experiments show that LXG proteins are exported out of *S. intermedius* cells and into adjacent competitor bacteria by the Esx pathway. Examining one of these LXG proteins in more detail showed that it can degrade a molecule that bacteria need to make their cell wall.

Together, these findings suggest that LXG proteins may influence the species living in many important microbial communities, including the human gut. Changes in the communities of gut microbes have been linked with many diseases. Therefore, understanding more about how the LXG proteins work may help us to develop ways to manipulate these communities to improve human health.

relatedness (*Hayes et al., 2014*). Additionally, both Proteobacteria and bacteria belonging to the divergent phylum Bacteroidetes deliver toxins to competitor Gram-negative cells in an indiscriminate fashion through the type VI secretion system (T6SS) (*Russell et al., 2014a*, *2014b*). Although toxin delivery by CDI and the T6SS is mechanistically distinct, cells harboring either pathway share the feature of prohibiting self-intoxication with immunity proteins that selectively inactivate cognate toxins through direct binding.

Few mechanisms that mediate direct antagonism between Gram-positive bacteria have been identified. In *Bacillus subtilis*, Sec-exported proteins belonging to the YD-repeat family have been shown to potently inhibit the growth of contacting cells belonging to the same strain (*Koskiniemi et al., 2013*); however, to our knowledge, a pathway that mediates interspecies antagonism between Gram-positive bacteria has not been identified. Given that Gram-positive and Gram-negative bacteria inhabit many of the same densely populated polymicrobial environments (e.g. the human gut), it stands to reason that the former should also possess mechanisms for more indiscriminate targeting of competing cells.

Contact-dependent toxin translocation between bacteria is primarily achieved using specialized secretion systems. Gram-negative export machineries of secretion types IV, V, and VI have each been implicated in this process (*Aoki et al., 2005*; *Hood et al., 2010*; *Souza et al., 2015*). A specialized secretion system widely distributed among Gram-positive bacteria is the Esx pathway (also referred to as type VII secretion) (*Abdallah et al., 2007*). This pathway was first identified in *Mycobacterium tuberculosis*, where it plays a critical role in virulence (*Stanley et al., 2003*). Indeed, attenuation of the vaccine strain *M. bovis* BCG can be attributed to a deletion inactivating ESX-1 secretion system present in virulence strains (*Lewis et al., 2003*; *Pym et al., 2003*). Subsequent genomic studies revealed that the Esx pathway is widely distributed in Actinobacteria, and that a divergent form is present in Firmicutes (*Gey Van Pittius et al., 2001*; *Pallen, 2002*). Though they share little genetic similarity, all Esx pathways studied to-date utilize a characteristic FtsK-like AAA+ ATPase referred to as EssC (or EccC) to catalyze the export of one or more substrates belonging to

the WXG100 protein family (*Ates et al., 2016*). Proteins in this family, including ESAT-6 (EsxA) and CFP10 (EsxB) from *M. tuberculosis*, heterodimerize in order to transit the secretion machinery.

The presence of the Esx secretion system in environmental bacteria as well as commensal and pathogenic bacteria that specialize in colonizing non-sterile sites of their hosts, suggests that the pathway may be functionally pliable. Supporting this notion, ESX-3 of *M. tuberculosis* is required for mycobactin siderophore-based iron acquisition and the ESX-1 and ESX-4 systems of *M. smegmatis* are linked to DNA transfer (*Gray et al., 2016*; *Siegrist et al., 2009*). In Firmicutes, a *Staphylococcus aureus* Esx-exported DNase toxin termed EssD (or EsaD) has been linked to virulence and contact-independent intraspecies antibacterial activity (*Cao et al., 2016*; *Ohr et al., 2017*).

Aravind and colleagues have noted that Esx secretion system genes are often linked to genes encoding polymorphic toxins belonging to the LXG protein family (*Zhang et al., 2012*). Analogous to characteristic antimicrobial polymorphic toxins of Gram-negative bacteria, the LXG proteins consist of a conserved N-terminal domain (LXG), a middle domain of variable length, and a C-terminal variable toxin domain. The LXG domain is predicted to adopt a structure resembling WXG100 proteins, thus leading to speculation that these proteins are Esx secretion system substrates (*Zhang et al., 2011*). Despite the association between LXG proteins and the Esx secretion system, to-date there are no experimental data linking them functionally. However, an intriguing study performed by Hayes and colleagues demonstrated antibacterial properties of *B. subtilis* LXG RNase toxins via heterologous expression in *E. coli* (*Holberger et al., 2012*). This growth inhibition was alleviated by co-expression of immunity determinants encoded adjacent to cognate LXG genes. We show here that LXG proteins transit the Esx secretion system of *Streptococcus intermedius* (*Si*) and function as antibacterial toxins that mediate contact-dependent interspecies antagonism.

## Results

### LXG proteins are Esx secretion system substrates

We initiated our investigation into the function of LXG proteins by characterizing the diversity and distribution of genes encoding these proteins across all sequenced genomes from Firmicutes. As noted previously, the C-terminal domains in the LXG family members we identified are highly divergent, exhibiting a wide range of predicted activities (*Figure 1a*) (*Zhang et al., 2012*). LXG protein-encoding genes are prevalent and broadly distributed in the classes Clostridiales, Bacillales and Lactobacillales (*Figure 1A*). Notably, a significant proportion of organisms in these taxa are specifically adapted to the mammalian gut environment. Indeed, we find that LXG genes derived from reference genomes of many of these gut-adapted bacteria are abundant in metagenomic datasets from human gut microbiome samples (*Figure 1A* and *Figure 1—figure supplement 1*). An LXG toxin that is predicted to possess ADP-ribosyltransferase activity – previously linked to interbacterial antagonism in Gram-negative organisms – was particularly abundant in a subset of human gut metagenomes (*Zhang et al., 2012*). Close homologs of this gene are found in *Ruminococcus*, a dominant taxa in the human gut microbiome, potentially explaining the frequency of this gene (*Wu et al., 2011*).

We next sought to determine whether LXG proteins are secreted via the Esx pathway. The toxin domain of several of the LXG proteins we identified shares homology and predicted catalytic residues with *M. tuberculosis* TNT, an $NAD^+$-degrading (NADase) enzyme (*Figure 2—figure supplement 1A*) (*Sun et al., 2015*). *Si*, a genetically tractable human commensal and opportunistic pathogen, is among the bacteria we identified that harbor a gene predicted to encode an NADase LXG protein (*Claridge et al., 2001*); we named this protein TelB (Toxin exported by Esx with LXG domain B). Attempts to clone the C-terminal toxin domain of TelB (TelB$_{tox}$) were initially unsuccessful, suggesting the protein exhibits a high degree of toxicity. Guided by the TNT structure, we circumvented this by assembling an attenuated variant (H661A) that was tolerated under non-induced conditions (TelB$_{tox}$*) (*Figure 2—figure supplement 1A*) (*Sun et al., 2015*). Induced expression of TelB$_{tox}$* inhibited *E. coli* growth and reduced cellular $NAD^+$ levels (*Figure 2A*, *Figure 2—figure supplement 1B*). The extent of $NAD^+$ depletion mirrored that catalyzed by expression of a previously characterized interbacterial NADase toxin, Tse6, and importantly, intracellular $NAD^+$ levels were unaffected by an unrelated bacteriostatic toxin, Tse2 (*Hood et al., 2010*; *Whitney et al., 2015*). Furthermore, substitution of a second predicted catalytic residue of TelB (R626A), abrogated toxicity of TelB$_{tox}$* and significantly restored $NAD^+$ levels (*Figure 2—figure supplement 1B–C*).

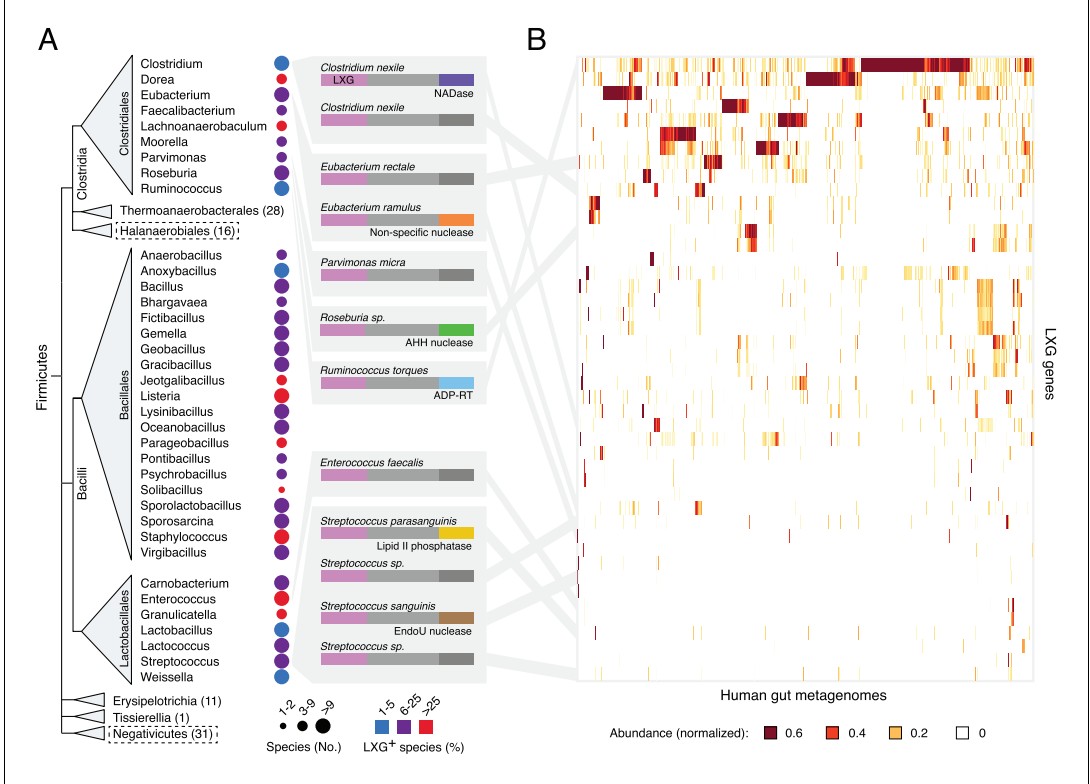

**Figure 1.** The LXG protein family contains diverse toxins that are broadly distributed in Firmicutes and found in the human gut microbiome. (**A**) Dendogram depicts LXG-containing genera within Firmicutes, clustered by class and order. Circle size indicates the number of sequenced genomes searched within each genus and circle color represents percentage of those found to contain at least one LXG protein. For classes or orders in which no LXG domain-containing proteins were found, the number of genera evaluated is indicated in parentheses; those consisting of Gram-negative organisms are boxed with dashed lines. Grey boxes contain predicted domain structures for representative divergent LXG proteins. Depicted are LXG-domains (pink), spacer regions (light grey) and C-terminal polymorphic toxin domains (NADase, purple; non-specific nuclease, orange; AHH family nuclease, green; ADP-ribosyltransferase, blue; lipid II phosphatase based on orthology to TelC (defined biochemically herein), yellow; EndoU family nuclease, brown; unknown activity, dark grey). (**B**) Heatmap depicting the relative abundance (using logarithmic scale) of selected LXG genes detected in the Integrated Gene Catalog (IGC). A complete heatmap is provided in *Figure 1—figure supplement 1*. Columns represent individual human gut metagenomes from the IGC database and rows correspond to LXG genes. Grey lines link representative LXG toxins in (**A**) to their corresponding (≥95% identity) IGC group in (**B**).

The following figure supplement is available for figure 1:

**Figure supplement 1.** Complete list of LXG genes found in human gut metagenomes.

Determination of the biochemical activity of TelB provided a means to test our hypothesis that LXG proteins are substrates of the ESX secretion pathway. Using an assay that exploits fluorescent derivatives of NAD$^+$ that form under strongly alkaline conditions, we found that concentrated cell-free supernatant of an *Si* strain containing *telB* (*Si*$^{B196}$) possesses elevated levels of NADase activity relative to that of a strain lacking *telB* (*Si*$^{27335}$) (*Figure 2B*) (*Johnson and Morrison, 1970*; *Olson et al., 2013*; *Whiley and Beighton, 1991*). Furthermore, the NADase activity present in the supernatant of *Si*$^{B196}$ was abolished by *telB* inactivation. Export of Esx substrates relies on EssC, a translocase with ATPase activity (*Burts et al., 2005*; *Rosenberg et al., 2015*). Inactivation of *essC* also abolished NADase activity in the supernatant of *Si*$^{B196}$, suggesting that TelB utilizes the Esx pathway for export.

The genome of *Si*$^{B196}$ encodes two additional LXG proteins, which we named TelA and TelC (*Figure 2C*). To determine if these proteins are also secreted in an Esx-dependent fashion, we collected cell-free supernatants from stationary phase cultures of wild-type and *essC*-deficient *Si*$^{B196}$. Extensive dialysis was used to reduce contamination from medium-derived peptides and the

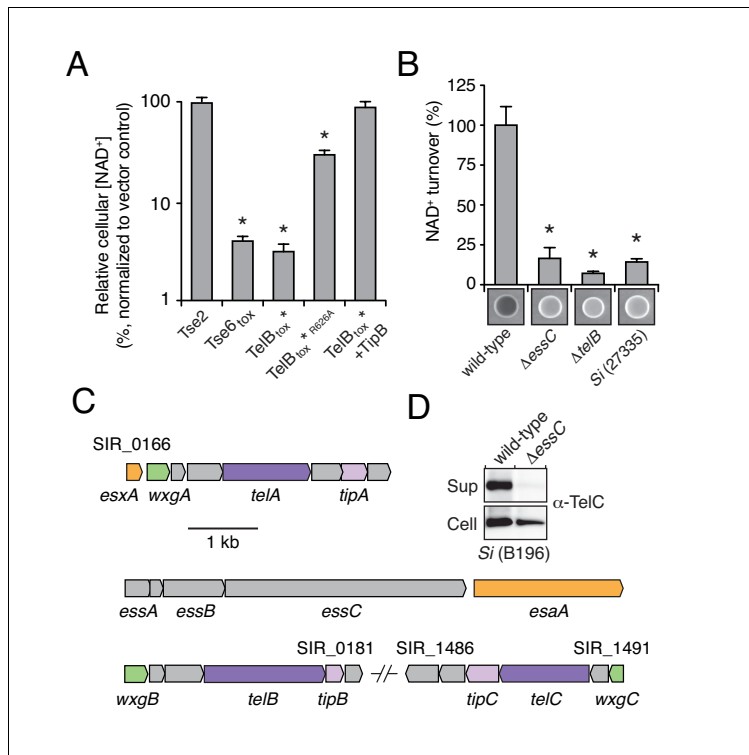

**Figure 2.** LXG-domain proteins of *S. intermedius* are secreted by the Esx-pathway. (**A**) NAD$^+$ levels in *E. coli* cells expressing a non-NAD$^+$-degrading toxin (Tse2), the toxin domain of a known NADase (Tse6$_{tox}$), an inducibly toxic variant of the C-terminal toxin domain of TelB (TelB$_{tox}$*), a variant of TelB$_{tox}$* with significantly reduced toxicity (TelB$_{tox}$*$^{R626A}$) and TelB$_{tox}$* co-expressed with its cognate immunity protein TipB. Cellular NAD$^+$ levels were assayed 60 min after induction of protein expression and were normalized to untreated cells. Mean values ($n$ = 3) ± SD are plotted. Asterisks indicate statistically significant differences in NAD$^+$ levels compared to vector control (p<0.05). (**B**) NAD$^+$ consumption by culture supernatants from the indicated *Si* strains. Fluorescent images of supernatant droplets supplemented with 2 mM NAD$^+$ for 3 hr; brightness is proportional to NAD$^+$ concentration and was quantified using densitometry. Mean values ± SD ($n$ = 3) are plotted. Asterisks indicate statistically significant differences in NAD$^+$ turnover compared to wild-type *Si*$^{B196}$ (p<0.05). (**C**) Regions of the *Si*$^{B196}$ genome encoding Esx-exported substrates. Genes are colored according to functions encoded (secreted Esx structural components, orange; secreted LXG toxins, dark purple; immunity determinants, light purple; WXG100-like proteins, green; other, grey). (**D**) Western blot analysis of TelC secretion in supernatant (Sup) and cell fractions of wild-type or *essC*-inactivated *Si*$^{B196}$.

The following figure supplement is available for figure 2:

**Figure supplement 1.** TelB resembles NADase toxins and inhibits the growth of bacteria.

remaining extracellular proteins were precipitated and identified using semi-quantitative mass spectrometry (*Liu et al., 2004*). This technique revealed that each of the LXG proteins predicted by the *Si* genome is exported in an Esx-dependent manner (*Table 1*). Western blot analysis of TelC secretion by wild-type and the *essC*-lacking mutant further validated Esx-dependent export (*Figure 2D*). Together, these data indicate that LXG proteins are substrates of the Esx secretion system.

## Contact-dependent interspecies antagonism is mediated by LXG toxins

The export of LXG proteins by the Esx pathway motivated us to investigate their capacity for mediating interbacterial antagonism. The C-terminal domains of TelA (TelA$_{tox}$) and TelC (TelC$_{tox}$) bear no homology to characterized proteins, so we first examined the ability of these domains to exhibit toxicity in bacteria. TelA$_{tox}$ and TelB$_{tox}$* inhibited growth when expressed in the cytoplasm of *E. coli*, whereas TelC$_{tox}$ did not exhibit toxicity in this cellular compartment (*Figure 3A*). Given the capacity

**Table 1.** The Esx-dependent extracellular proteome of *S. intermedius* B196.

| Locus tag | Wild-type | ΔessC | Relative abundance (Wild-type/ΔessC) | Esx function | Name |
|---|---|---|---|---|---|
| SIR_0169* | 19.67[†] | 0 | Not detected in ΔessC | LXG protein[‡] | TelA |
| SIR_0176 | 14.67 | 0 | Not detected in ΔessC | Structural component | EsaA |
| SIR_1489 | 12.00 | 0 | Not detected in ΔessC | LXG protein | TelC |
| SIR_1516 | 9.33 | 0 | Not detected in ΔessC | - | Trigger Factor |
| SIR_0179 | 5.33 | 0 | Not detected in ΔessC | LXG protein | TelB |
| SIR_0166 | 140.00 | 17.48 | 8.01 | Structural component | EsxA |
| SIR_0273 | 15.33 | 2.28 | 6.73 | - | - |
| SIR_1626 | 15.00 | 2.28 | 6.58 | - | GroEL |
| SIR_0832 | 12.33 | 8.36 | 1.48 | - | Enolase |
| SIR_1904 | 49.00 | 37.24 | 1.32 | - | Putative serine protease |
| SIR_1382 | 26.00 | 19.76 | 1.32 | - | Fructose-bisphosphate aldolase |
| SIR_0648 | 21.67 | 17.48 | 1.24 | - | 50S ribosomal protein L7/L12 |
| SIR_0212 | 47.00 | 39.52 | 1.19 | - | Elongation Factor G |
| SIR_0081 | 8.67 | 7.60 | 1.14 | - | Putative outer membrane protein |
| SIR_1676 | 16.33 | 14.44 | 1.13 | - | phosphoglycerate kinase |
| SIR_1523 | 12.67 | 12.92 | 0.98 | - | DnaK |
| SIR_1154 | 10.33 | 10.64 | 0.97 | - | Putative bacteriocin accessory protein |
| SIR_1027 | 63.00 | 67.64 | 0.93 | - | Elongation Factor Tu |
| SIR_1455 | 14.00 | 15.96 | 0.88 | - | - |
| SIR_0758 | 13.00 | 15.20 | 0.86 | - | - |
| SIR_1387 | 9.33 | 11.40 | 0.82 | - | Putative extracellular solute-binding protein |
| SIR_0492 | 12.33 | 15.20 | 0.81 | - | Putative adhesion protein |
| SIR_1033 | 17.67 | 24.32 | 0.73 | - | - |
| SIR_1359 | 14.00 | 19.76 | 0.71 | - | Penicillin-binding protein 3 |
| SIR_0011 | 12.33 | 17.48 | 0.71 | - | Beta-lactamase class A |
| SIR_1546 | 8.33 | 12.16 | 0.69 | - | - |
| SIR_0040 | 101.67 | 160.36 | 0.63 | - | Putative stress protein |
| SIR_1608 | 11.00 | 18.24 | 0.60 | - | Putative endopeptidase O |
| SIR_1549 | 7.33 | 12.16 | 0.60 | - | - |
| SIR_1675 | 79.00 | 132.24 | 0.60 | - | Putative cell-surface antigen I/II |
| SIR_1418 | 11.33 | 21.28 | 0.53 | - | Putative transcriptional regulator LytR |
| SIR_0080 | 11.00 | 21.28 | 0.52 | - | - |
| SIR_1025 | 28.33 | 63.84 | 0.44 | - | Lysozyme |
| SIR_0113 | 10.67 | 24.32 | 0.44 | - | - |
| SIR_0297 | 8.33 | 24.32 | 0.34 | - | - |

*Rows highlighted in green correspond to proteins linked to the Esx pathway.

[†]Values correspond to average SC (spectral counts) of triplicate biological replicates for each strain.

[‡]Functional link of LXG proteins to Esx secretion pathway defined in the study.

of some interbacterial toxins to act on extracellular structures, we assessed the viability of *Si* cells expressing TelC$_{tox}$ targeted to the sec translocon. In contrast to TelC$_{tox}$ production, overexpression of a derivative bearing a signal peptide directing extracellular expression (ss-TelC$_{tox}$) exhibited significant toxicity (*Figure 3B*).

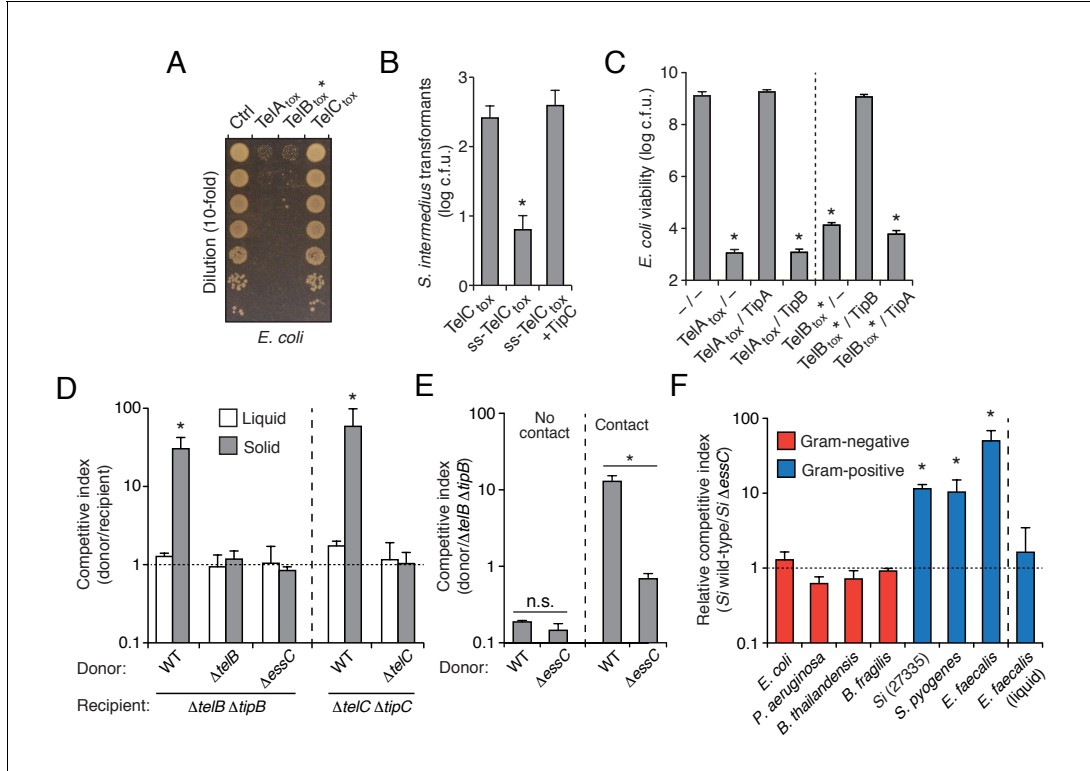

**Figure 3.** *S. intermedius* LXG proteins inhibit bacterial growth and mediate contact-dependent interbacterial antagonism. (**A**) Viability of *E. coli* cells grown on solid media harboring inducible plasmids expressing the C-terminal toxin domains of the three identified $Si^{B196}$ LXG proteins or an empty vector control. (**B**) $Si^{B196}$ colonies recovered after transformation with equal concentrations of constitutive expression plasmids carrying genes encoding the indicated proteins. ss-TelC$_{tox}$ is targeted to the sec translocon through the addition of the secretion signal sequence from *S. pneumoniae* LysM (SP_0107). Error bars represent ± SD (*n* = 3). Asterisk indicates a statistically significant difference in *Si* transformation efficiency relative to TelC$_{tox}$ (p<0.05). (**C**) Viability of *E. coli* cells grown on solid media harboring inducible plasmids co-expressing the indicated proteins. Empty vector controls are indicated by a dash. Mean c.f.u. values ± SD (*n* = 3) are plotted. Asterisks indicate statistically significant differences in *E. coli* viability relative to vector control (p<0.05) (**D**) Intra-species growth competition experiments between the indicated bacterial strains. Competing strains were mixed and incubated in liquid medium or on solid medium for 30 hr and both initial and final populations of each strain were enumerated by plating on selective media. The competitive index was determined by comparing final and initial ratios of the two strains. Asterisks indicate outcomes statistically different between liquid and solid medium (*n* = 3, p<0.05). (**E**) Intra-species growth competition experiments performed as in (**D**) except for the presence of a filter that inhibits cell-cell contact. No contact, filter placed between indicated donor and susceptible recipient (Δ*telB* Δ*tipB*) strains; Contact, donor and susceptible recipient strains mixed on same side of filter. Asterisks indicate statistically different outcomes (*n* = 3, p<0.05). Note that recipient cell populations have an Esx-independent fitness advantage in these experiments by virtue of their relative proximity to the growth substrate. (**F**) Inter-species growth competition experiments performed on solid or in liquid (*E. faecalis*) medium between *Si* wild-type and Δ*essC* donor strains and the indicated recipient organisms. $Si^{23775}$ lacks *tipA* and *tipB* and is therefore potentially susceptible to TelA and TelB delivered by $Si^{B196}$. Asterisks indicate outcomes where the competitive index of wild-type was significantly higher than an Δ*essC* donor strain (*n* = 3, p<0.05). Genetic complementation of the mutant phenotypes presented in this figure was confounded by inherent plasmid fitness costs irrespective of the inserted sequence. As an alternative, we performed whole genome sequencing on strains Δ*essC*, Δ*telB*, Δ*telC*, Δ*telB* Δ*tipB*, and Δ*telC* Δ*tipC*, which confirmed the respective desired mutation as the only genetic difference between these strains. Sequences of these strains have been deposited to the NCBI Sequence Read Archive (BioProject ID: PRJNA388094).

The following figure supplements are available for figure 3:

**Figure supplement 1.** TelC directly interacts with its cognate immunity protein TipC.

**Figure supplement 2.** TelC levels elevated by high cell density or addition of purified protein fail to yield cellular intoxication in liquid media.

We next evaluated whether the Tel proteins, like the substrates of interbacterial toxin delivery systems in Gram-negative bacteria, are inactivated by genetically linked specialized cognate immunity determinants. By co-expressing candidate open reading frames located downstream of each *tel* gene, we identified a cognate *tip* (tel immunity protein) for each toxin (*Figure 3B–C* and *Figure 3— figure supplement 1*). We then sought to inactivate each of these factors to generate $Si^{B196}$ strains sensitive to each of the Tel proteins. In $Si^{B196}$, *telA tipA* loci are located immediately upstream of conserved *esx* genes (*Figure 2C*). We were unable to generate non-polar *telA tipA*-inactivated strains, and thus focused our efforts on the other two *tel tip* loci.

We reasoned that if LXG toxins target non-self cells, this process would occur either through diffusion or by facilitated transfer, the latter of which would likely require cell contact. Since we detect TelA-C secretion in liquid medium, we began our attempts to observe intercellular intoxication with wild-type and toxin-sensitive target cell co-culture. These efforts yielded no evidence of target cell killing or growth inhibition, including when co-incubations were performed at cell densities higher than that achievable through growth (*Figure 3D*, *Figure 3—figure supplement 2A*). The application of concentrated supernatants or purified TelC (to a final concentration of 0.1 mg/mL) to sensitive strains also did not produce evidence of toxicity (*Figure 3—figure supplement 2B–C*). This result is perhaps not surprising given the barrier presented by the Gram-positive cell wall (*Forster and Marquis, 2012*).

Next, we tested conditions that enforce cell contact. In each of these experiments, donor and recipient strains were grown in pure culture before they were mixed at defined ratios and cultured on a solid surface for 30 hr to promote cell-cell interactions. We observed significant growth inhibition of TelB- or TelC-susceptible strains co-cultured with wild-type, but not when co-cultured with strains lacking *telB* or *telC*, respectively (*Figure 3D*). A strain bearing inactivated *essC* was also unable to intoxicate a sensitive recipient. In competition experiments performed in parallel wherein the bacterial mixtures were grown in liquid culture, TelB and TelC-susceptible strains competed equally with wild type, suggesting that Esx-mediated intoxication requires prolonged cell contact. To further probe this requirement, we conducted related experiments in which wild-type donor cells were segregated from sensitive recipients by a semi-permeable (0.2 μm pore size) membrane (*Figure 3E*). This physical separation blocked intoxication, which taken together with the results of our liquid co-culture experiments and our finding that purified TelC is not bactericidal, strongly suggests that the mechanism of Esx-dependent intercellular LXG protein delivery requires immediate cell-cell contact.

In Gram-negative bacteria, some antagonistic cell contact-dependent pathways display narrow target range, whereas others act between species, or even between phyla (*Hayes et al., 2014*; *Russell et al., 2014a*). To begin to determine the target range of Esx-based LXG protein delivery, we measured its contribution to $Si^{B196}$ fitness in interbacterial competition experiments with a panel of Gram-positive and -negative bacteria. The Esx pathway conferred fitness to $Si^{B196}$ in competition with $Si^{23775}$, *S. pyogenes*, and *Enterococcus faecalis*, an organism from a closely related genus (*Figure 3F*). On the contrary, the pathway did not measurably affect the competitiveness of $Si^{B196}$ against Gram-negative species belonging to the phyla Proteobacteria (*E. coli, Burkholderia thailandensis, Pseudomonas aeruginosa*) or Bacteroidetes (*Bacteriodes fragilis*). These results demonstrate that the Esx pathway can act between species and suggest that its target range may be limited to Gram-positive bacteria.

## TelC targets the bacterial cell wall biosynthetic precursor lipid II

The Esx pathway is best known for its role in mediating pathogen-host cell interactions (*Abdallah et al., 2007*). Given this precedence, we considered the possibility that the antibacterial activity we observed may not be relevant physiologically. TelB degrades $NAD^+$, a molecule essential for all cellular life, and therefore this toxin is not definitive in this regard. We next turned our attention to TelC, which elicits toxicity from outside of the bacterial cell (*Figure 3B*). This protein contains a conserved aspartate-rich motif that we hypothesized constitutes its enzymatic active site (*Figure 4—figure supplement 1A*). To gain further insight into TelC function, we determined the crystal structure of $TelC_{tox}$ to 2.0 Å resolution (*Table 2*). The structure of $TelC_{tox}$ represents a new fold; it is comprised of distinct and largely α-helical N- and C-terminal lobes (*Figure 4A*). The single β element of $TelC_{tox}$ is a hairpin that protrudes from the N-terminal lobe. Although $TelC_{tox}$ does not share significant similarity to previously determined structures, we located its putative active site within a

**Table 2.** X-ray data collection and refinement statistics.

| | $TelC_{202-CT}$ (Semet) |
|---|---|
| **Data Collection** | |
| Wavelength (Å) | 0.979 |
| Space group | $C222_1$ |
| Cell dimensions | |
| a, b, c (Å) | 127.4, 132.7, 58.3 |
| α, β, γ (°) | 90.0, 90.0, 90.0 |
| Resolution (Å) | 49.20–1.98 (2.03–1.98)* |
| Total observations | 891817 |
| Unique observations | 34824 |
| $R_{pim}$ (%) | 6.6 (138.5) |
| $I/\sigma I$ | 11.4 (0.8) |
| Completeness (%) | 100.0 (99.9) |
| Redundancy | 25.6 (23.4) |
| **Refinement** | |
| $R_{work}$ / $R_{free}$ (%) | 22.4/24.6 |
| Average B-factors (Å$^2$) | 53.8 |
| No. atoms | |
| Protein | 2539 |
| Ligands | 3 |
| Water | 145 |
| Rms deviations | |
| Bond lengths (Å) | 0.008 |
| Bond angles (°) | 0.884 |
| Ramachandran plot (%) | |
| Total favored | 96.9 |
| Total allowed | 99.7 |
| Coordinate error (Å) | 0.28 |
| PDB code | 5UKH |

*Values in parentheses correspond to the highest resolution shell.

shallow groove that separates the N- and C-terminal lobes. This region contains a calcium ion bound to several residues that comprise the conserved aspartate-rich motif. Site-specific mutagenesis of these residues abrogated TelC-based toxicity (*Figure 4B,C*, *Figure 4—figure supplement 1B*).

We next assessed the morphology of cells undergoing intoxication by $TelC_{tox}$. Due to the potent toxicity of $TelC_{tox}$ in *Si*, we employed an inducible expression system in *S. aureus* as an alternative. *S. aureus* cells expressing extracellularly-targeted $TelC_{tox}$ exhibited significantly reduced viability (*Figure 4D*), and when examined microscopically, displayed a cessation of cell growth followed by lysis that was not observed in control cells (*Figure 4E*, *Videos 1–2*). Despite eliciting effects consistent with cell wall peptidoglycan disruption, isolated cell walls treated with $TelC_{tox}$ and peptidoglycan recovered from cells undergoing TelC-based intoxication showed no evidence of enzymatic digestion (*Figure 4—figure supplement 2A–D*). These data prompted us to consider that TelC corrupts peptidoglycan biosynthesis, which could also lead to the lytic phenotype observed (*Harkness and Braun, 1989*).

The immediate precursor of peptidoglycan is lipid II, which consists of the oligopeptide disaccharide repeat unit linked via pyrophosphate to a lipid carrier (*Vollmer and Bertsche, 2008*). Likely due to its distinctive and conserved structure, lipid II is the target of diverse antibacterial molecules

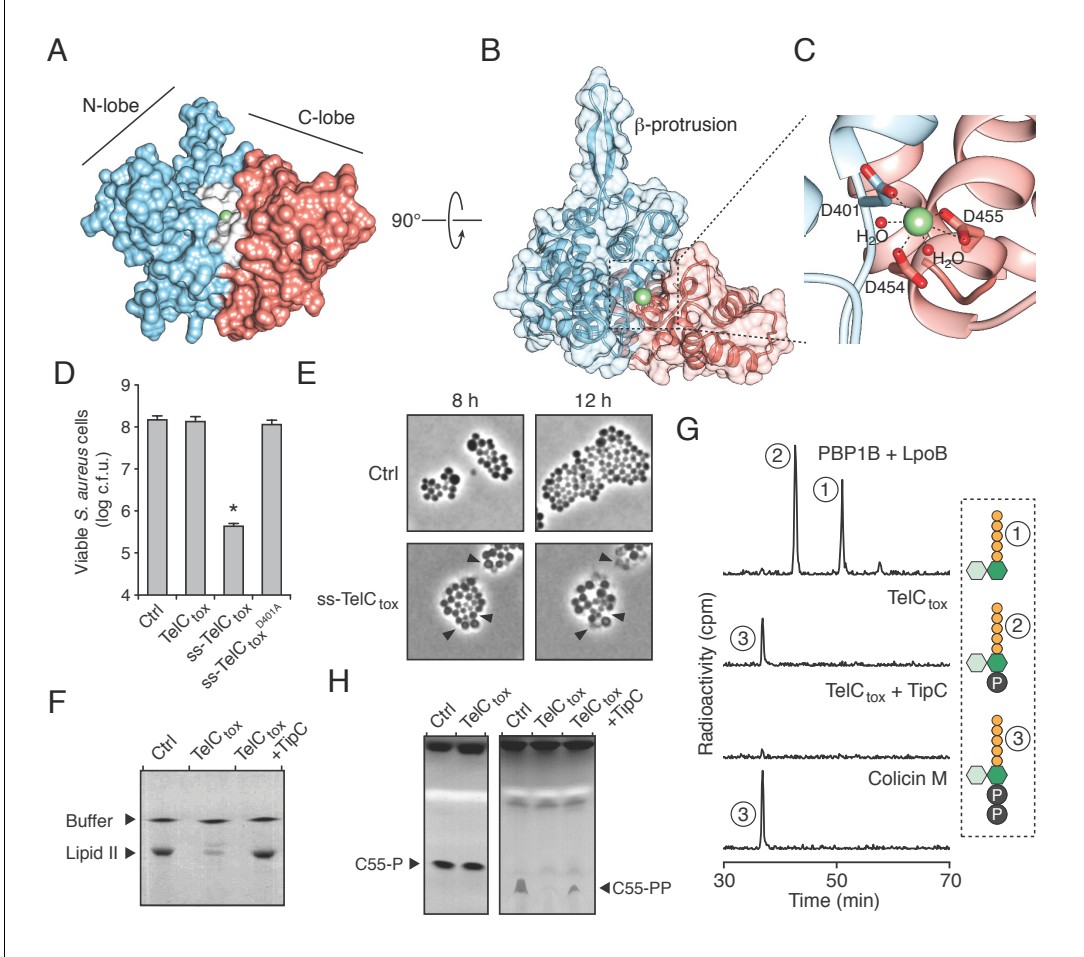

**Figure 4.** TelC is a calcium-dependent lipid II phosphatase. (**A**) Space-filling representation of the 2.0 Å resolution TelC$_{tox}$ X-ray crystal structure. Protein lobes (red and blue), active site cleft (white) and Ca$^{2+}$ (green) are indicated. (**B**) TelC$_{tox}$ structure rotated as indicated relative to (**A**) with transparent surface revealing secondary structure. (**C**) Magnification of the TelC active site showing Ca$^{2+}$ coordination by conserved aspartate residues and water molecules. (**D**) Viability of *S. aureus* cells harboring inducible plasmids expressing the indicated proteins or a vector control. ss-TelC$_{tox}$ is targeted for secretion through the addition of the signal sequence encoded by the 5' end of the *hla* gene from *S. aureus*. Mean c.f.u. values ± SD (*n* = 3) are plotted. Asterisk indicates a statistically significant difference in *S. aureus* viability relative to vector control (p<0.05) (**E**) Representative micrographs of *S. aureus* expressing ss-TelC$_{tox}$ or a vector control. Frames were acquired eight and 12 hr after spotting cells on inducing growth media. (**F**) Thin-layer chromotography (TLC) analysis of reaction products from incubation of synthetic Lys-type lipid II with buffer (Ctrl), TelC$_{tox}$, or TelC$_{tox}$ and its cognate immunity protein TipC. (**G**) Partial HPLC chromatograms of radiolabeled peptidoglycan (PG) fragments released upon incubation of Lys-type lipid II with the indicated purified proteins. Schematics depict PG fragment structures (pentapeptide, orange; N-acetylmuramic acid, dark green; N-acetylglucosamine, light green; phosphate, black). Known fragment patterns generated by PBP1B + LpoB and colicin M serve as controls. (**H**) TLC analysis of reaction products generated from incubation of buffer (Ctrl), TelC$_{tox}$ or TelC$_{tox}$ and TipC with undecaprenyl phosphate (C55–P) (left) or undecaprenyl pyrophosphate (C55–PP) (right).

The following figure supplements are available for figure 4:

**Figure supplement 1.** TelC contains an aspartate-rich motif required for toxicity.

**Figure supplement 2.** TelC does not degrade intact Gram-positive sacculi.

**Figure supplement 3.** TelC degrades lipid II, contributes to interbacterial antagonism and is not toxic to yeast cells.

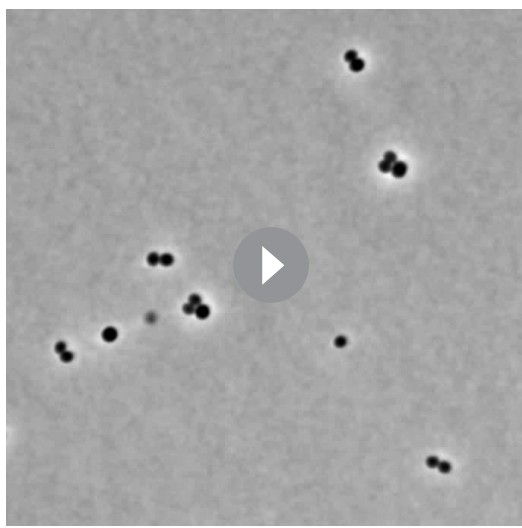

**Video 1.** Time-lapse series of *S. aureus* USA300 pEPSA5 growth. Cells were imaged every 10 min.

(*Breukink and de Kruijff, 2006*; *Oppedijk et al., 2016*). To test activity against lipid II, we incubated the molecule with purified TelC$_{tox}$. Analysis of the reaction products showed that TelC$_{tox}$ cleaves lipid II – severing the molecule at the phosphoester linkage to undecaprenyl (*Figure 4F–G*, *Figure 4—figure supplement 3A*). Reaction products were confirmed by mass spectrometry and inclusion of TipC inhibited their formation. Consumption of lipid II for peptidoglycan assembly generates undecaprenyl pyrophosphate (UPP), which is converted to undecaprenyl phosphate (UP), and transported inside the cell. The UP molecule then reenters peptidoglycan biosynthesis or is utilized as a carrier for another essential cell wall constituent, wall teichoic acid (WTA). Our experiments showed that TelC$_{tox}$ is capable of hydrolyzing cleaved undecaprenyl derivatives but displays a strict requirement for the pyrophosphate group (*Figure 4H*), indicating the potential for TelC to simultaneously disrupt two critical Gram-positive cell wall polymers. Consistent with its ability to inhibit a conserved step in peptidoglycan biosynthesis, TelC exhibited toxicity towards diverse Gram-positive species including *Si* (*Figure 2B*), *S. aureus* (*Figure 4D*) and *E. faecalis* (*Figure 4—figure supplement 3B*). These data do not explain our observation that cytoplasmic TelC is non-toxic, as the substrates we defined are present in this compartment. The substrates may be inaccessible or TelC could be inactive in the cytoplasm. It is worth noting that TelC contains a calcium ion bound at the interface of its N- and C-terminal lobes. Many secreted proteins that bind calcium utilize the abundance of the free ion in the milieu to catalyze folding. Taken together, our biochemical and phenotypic data strongly suggest that TelC is a toxin directed specifically against bacteria. While we cannot rule out that TelC may have other targets, we find that its expression in the cytoplasm or secretory pathway of yeast does not impact the viability of this model eukaryotic cell (*Figure 4—figure supplement 3C–D*).

## WXG100-like proteins bind cognate LXG proteins and promote toxin export

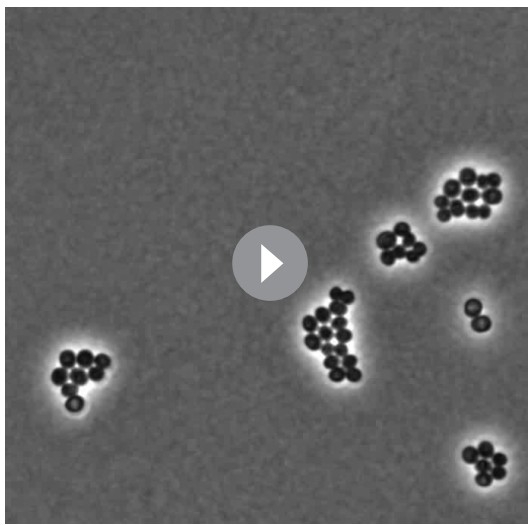

**Video 2.** Time-lapse series of *S. aureus* USA300 pEPSA5::ss-telC$_{tox}$ growth. Cells were imaged every 10 min.

The majority of Esx substrates identified to-date belong to the WXG100 protein family. These proteins typically display secretion co-dependency and are essential for apparatus function. *M. tuberculosis* ESX-1 exports two WXG100 proteins, ESAT-6 and CFP10, and the removal of either inhibits the export of other substrates (*Ates et al., 2016*; *Renshaw et al., 2002*). LXG proteins do not belong to the WXG100 family; thus, we sought to determine how the Tel proteins influence Esx function in *Si*. Using Western blot analysis to measure TelC secretion and extracellular NADase activity as a proxy for TelB secretion, we found that *telB*- and *telC*-inactivated strains of *Si* retain the capacity to secrete TelC and TelB, respectively (*Figure 5A–B*). These data indicate that TelB and TelC are not required for core apparatus function and do not display secretion co-dependency.

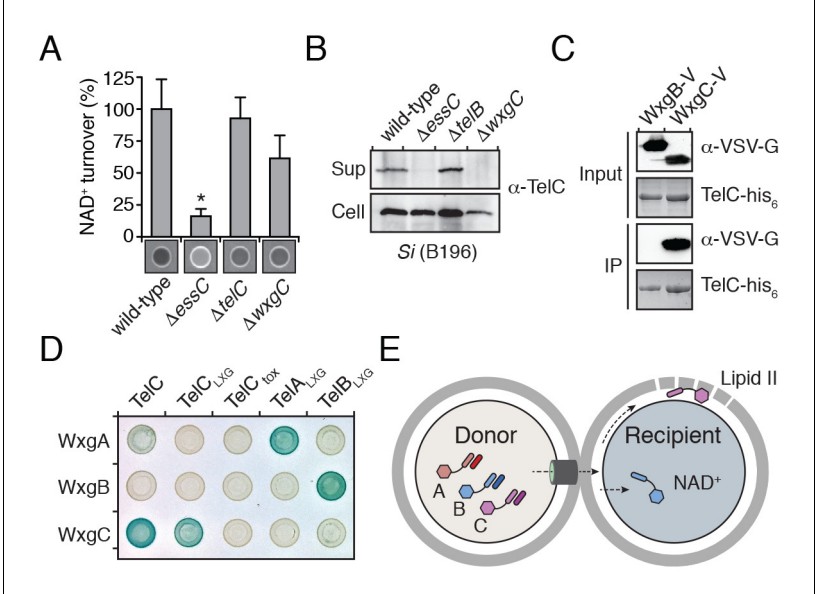

**Figure 5.** LXG domain proteins are independently secreted and require interaction with cognate WXG100-like partners for export. (**A**) NAD$^+$ consumption assay of culture supernatants of the indicated $Si^{B196}$ strains. Mean densitometry values ± SD ($n = 3$) are plotted. Asterisk indicates statistically significant difference in NAD$^+$ turnover compared to wild-type $Si^{B196}$ ($p<0.05$). (**B**) Western blot analysis of TelC secretion in supernatant (Sup) and cell fractions. (**C**) Western blot and coomassie stain analysis of CoIP assays of TelC-his$_6$ co-expressed with either WxgB-V or WxgC-V proteins. (**D**) Bacterial two-hybrid assay for interaction between Tel and WXG100-like proteins. Adenylate cyclase subunit T25 fusions (WXG100-like proteins) and T18 fusions (Tel proteins and fragments thereof) were co-expressed in the indicated combinations. Bait-prey interaction results in blue color production. (**E**) Model depicting Esx-dependent cell-cell delivery of LXG toxins between bacteria. The schematic shows an $Si$ donor cell containing cognate TelA-C (light shades) and WxgA-C (dark shades) pairs intoxicating a susceptible recipient cell. Molecular targets of LXG toxins identified in this study are depicted in the recipient cell.

The following figure supplement is available for figure 5:

**Figure supplement 1.** Domain architecture of the Tel proteins.

---

Interestingly, we noted genes encoding WXG100-like proteins upstream of *telA-C* (*wxgA-C*) (**Figure 2C**); however, these proteins were not identified in the extracellular proteome of *Si* (**Table 1**). Given the propensity for Esx substrates to function as heterodimers, we hypothesized that the Tel proteins specifically interact with cognate Wxg partners. In support of this, we found that WxgC, but not WxgB co-purified with TelC (**Figure 5C**). Moreover, using bacterial two-hybrid assays, we determined that this interaction is mediated by the LXG domain of TelC (**Figure 5D**). To investigate the generality of these findings, we next examined all pairwise interactions between the three Wxg proteins and the LXG domains of the three Tel proteins (TelA-C$_{LXG}$) (**Figure 5—figure supplement 1**). We found that WxgA-C interact specifically with the LXG domain of their cognate toxins (**Figure 5D**). The functional relevance of the LXG–WXG100 interaction was tested by examining substrate secretion in a strain lacking *wxgC*. We found that *wxgC* inactivation abrogates TelC secretion, but not that of TelB (**Figure 5A–B**). In summary, these data suggest that cognate Tel–Wxg interaction facilitates secretion through the Esx pathway of *Si* (**Figure 5E**).

## Discussion

We present multiple lines of evidence that Esx-mediated delivery of LXG toxins serves as a physiological mechanism for interbacterial antagonism between Gram-positive bacteria. Our results suggest that like the T6S pathway of Gram-negative bacteria, the Esx system may mediate antagonism against diverse targets, ranging from related strains to species belonging to other genera

(*Schwarz et al., 2010*). This feature of Esx secretion, in conjunction with the frequency by which we detect LXG genes in human gut metagenomes, suggests that the system could have significant ramifications for the composition of human-associated polymicrobial communities. Bacteria harboring LXG toxin genes are also components or pathogenic invaders of polymicrobial communities important in agriculture and food processing. For instance, LXG toxins may assist *Listeria* in colonizing fermented food communities dominated by *Lactobacillus* and *Lactococcus* (*Farber and Peterkin, 1991*). Of note, the latter genera also possess LXG toxins, which may augment their known antimicrobial properties. Our findings thus provide insights into the forces influencing the formation of diverse communities relevant to human health and industry.

Palmer and colleagues recently reported that the Esx system of *Staphylococcus aureus* exports EssD, a nuclease capable of inhibiting the growth of target bacteria in co-culture (*Cao et al., 2016*). The relationship between these findings and those we report herein is currently unclear. *S. aureus* EssD does not possess an LXG domain and was reported to be active against susceptible bacteria during co-incubation in liquid media, a condition we found not conducive to LXG toxin delivery (*Figure 3D*). It is evident that the Esx pathway is functionally pliable (*Burts et al., 2005*; *Conrad et al., 2017*; *Gray et al., 2016*; *Gröschel et al., 2016*; *Manzanillo et al., 2012*; *Siegrist et al., 2009*); therefore, it is conceivable that it targets toxins to bacteria through multiple mechanisms. The capacity of EssD to act against bacteria in liquid media could be the result of its over-expression from a plasmid, although we found that the exogenous administration of quantities of TelC far exceeding those likely achievable physiologically had no impact on sensitive recipient cells (*Figure 3—figure supplement 2C*). A later study of EssD function found no evidence of inter-bacterial targeting and instead reported that its nuclease activity affects IL-12 accumulation in infected mice (*Ohr et al., 2017*).

Our data suggest that, like a subset of substrates of the Esx systems of *M. tuberculosis*, LXG family members require hetero-dimerization with specific WXG100-like partners to be secreted (*Ates et al., 2016*). Hetero-dimerization is thought to facilitate secretion of these substrates due to the requirement for a bipartite secretion signal consisting of a YxxxD/E motif in the C-terminus of one partner in proximity to the WXG motif present in the turn between helices in the second protein (*Champion et al., 2006*; *Daleke et al., 2012a*; *Poulsen et al., 2014*; *Sysoeva et al., 2014*). While the canonical secretion signals found in other Esx substrates appear to be lacking in the LXG proteins and their interaction partners, structure prediction algorithms suggest they adopt similar helical hairpin structures, which could facilitate formation of an alternative form of the bipartite signal. Unlike previously characterized Esx substrates, we found that the LXG proteins are not co-dependent for secretion, and we failed to detect secretion of their WXG100-like interaction partners. This suggests that WxgA-C could function analogously to the EspG proteins of *M. tuberculosis*, which serve as intracellular chaperones facilitating delivery of specific substrates to the secretion machinery (*Daleke et al., 2012b*; *Ekiert and Cox, 2014*). Alternatively, Wxg–Lxg complexes could be secreted as heterodimers, but for technical reasons the Wxg member was undetected in our experiments. The paradigm of Lxg-Wxg interaction likely extends beyond *S. intermedius*, as we observe that LXG proteins from other species are commonly encoded within the same operon as Wxg homologs.

Our study leaves open the question of how Esx-exported LXG proteins reach their targets. In the case of TelC, the target resides on the extracellular face of the plasma membrane, and in the case of TelA and TelB, they are cytoplasmic. Crossing the thick Gram-positive cell wall is the first hurdle that must be overcome to deliver of each of these toxins. The size of LXG toxins exceeds that of molecules capable of free diffusion across the peptidoglycan sacculus (*Forster and Marquis, 2012*). Donor cell-derived cell wall hydrolytic enzymes may facilitate entry or the LXG proteins could exploit cell surface proteins present on recipient cells. Whether the entry of LXG toxins is directly coordinated by the Esx pathway is not known; our experiments do not rule-out that the requirement for donor-recipient cell contact reflects a step subsequent to secretion by the Esx pathway. Once beyond the sacculus, TelA and TelB must translocate across the plasma membrane. Our study has identified roles for the N- and C-terminal domains of LXG proteins; however, the function of the region between these two domains remains undefined and may participate in entry. Intriguingly, the central domains of TelA and TelB are each over 150 residues, whereas the LXG and toxin domains of TelC, which does not require access to the cytoplasm, appear to directly fuse (*Figure 5—figure supplement 1*). Based on the entry mechanisms employed by other interbacterial toxins, this central domain – or another part of the protein – could facilitate direct translocation, proteolytic release of

the toxin domain, interaction with a recipient membrane protein, or a combination of these activities (*Kleanthous, 2010*; *Willett et al., 2015*).

We discovered that TelC, a protein lacking characterized homologs, adopts a previously unobserved fold and catalyzes degradation of the cell wall precursor molecule lipid II. This molecule is the target of the food preservative nisin, as well as the last-line antibiotic vancomycin, which is used to treat a variety of Gram-positive infections (*Ng and Chan, 2016*). Lipid II is also the target of the recently discovered antibiotic teixobactin, synthesized by the soil bacterium *Eleftheria terrae* (*Ling et al., 2015*). A particularly interesting property of this potential therapeutic is the low rate at which resistance is evolved. The apparent challenge of structurally modifying lipid II in order to subvert antimicrobials may explain why interbacterial toxins targeting this molecule have evolved independently in Gram-negative (colicin M) and -positive (TelC) bacteria (*El Ghachi et al., 2006*). We anticipate that biochemical characterization of additional LXG toxins of unknown function will reveal further Gram-positive cell vulnerabilities that could likewise be exploited in the design of new antibiotics.

## Materials and methods

### Bacterial strains and growth conditions

*S. intermedius* strains used in this study were derived from the sequenced strains ATCC 27335 and B196 (*Supplementary file 1*). *S. intermedius* strains were grown at 37°C in the presence of 5% $CO_2$ in Todd Hewitt broth (THYB) or agar (THYA) supplemented with 0.5% yeast extract. When needed, media contained spectinomycin (75 µg/mL) or kanamycin (250 µg/mL). *S. aureus* USA300 derived strains were grown at 37°C in tryptic soy broth (TSB) or agar (TSA) supplemented with chloramphenicol (10 µg/mL) and xylose (2% *w/v*) when needed. *E. faecalis* OG1RF and *S. pyogenes* 5005 were grown at 37°C on Brain Heart Infusion (BHI) media. *P. aeruginosa* PAO1 and *B. thailandensis* E264 were grown at 37°C on THYA. *B. fragilis* NCTC9343 was grown anaerobically at 37°C on Brain Heart Infusion-supplemented (BHIS) media. *E. coli* strains used in this study included DH5α for plasmid maintenance, BL21 for protein expression and toxicity assays and MG1655 for competition experiments. *E. coli* strains were grown on LB medium supplemented with 150 µg/mL carbenicillin, 50 µg/mL kanamycin, 200 µg/mL trimethoprim, 75 µg/mL spectinomycin, 200 µM IPTG or 0.1% (*w/v*) rhamnose as needed. For co-culture experiments with *S. intermedius* strains, *E. coli*, *B. thailandensis*, *P. aeruginosa*, *S. aureus*, *E. faecalis*, *S. pyogenes* were grown on THYA. BHIS agar supplemented with sheep's blood was used when *B. fragilis* was grown in co-culture with *S. intermedius*. *S. cerevisiae* BY4742 was grown on Synthetic Complete -uracil (SC-ura) medium at 30°C.

*S. intermedius* mutants were generated by replacing the gene to be deleted with a cassette conferring resistance to spectinomycin (derived from pDL277) or kanamycin (derived from pBAV1K-T5), as previously described (*Tomoyasu et al., 2010*). Briefly, the antibiotic resistance cassette was cloned between ~800 bp of sequence homologous to the regions flanking the gene to be deleted. The DNA fragment containing the cassette and flanking sequences was then linearized by restriction digest, gel purified, and ~250 ng of the purified fragment was added to 2 mL of log-phase culture pre-treated for two hours with competence peptide (200 ng/ml) to stimulate natural transformation. Cultures were further grown for four hours before plating on the appropriate antibiotic. All deletions were confirmed by PCR.

### DNA manipulation and plasmid construction

All DNA manipulation procedures followed standard molecular biology protocols. Primers were synthesized and purified by Integrated DNA Technologies (IDT). Phusion polymerase, restriction enzymes and T4 DNA ligase were obtained from New England Biolabs (NEB). DNA sequencing was performed by Genewiz Incorporated.

### Informatic analysis of LXG protein distribution

A comprehensive list of all clade names in the Firmicutes phylum was obtained from the List of Prokaryotic names with Standing in Nomenclature (http://www.bacterio.net/; updated 2017-02-02), a database that compiles comprehensive journal citations for every characterized prokaryotic species (*Euzéby, 1997*). This list was then compared with results obtained from a manually curated

Jackhmmer search and LXG-containing Firmicutes were tabulated at the order, family, and genus levels (*Finn et al., 2015*; *Mitchell et al., 2015*). These results were binned into three categories based on the number of sequenced species and then further differentiated by the number of LXG-positive species within each genus. For species belonging to orders containing no predicted LXG encoding genes, the number of genera examined was tabulated and included in the dendogram.

## Identification of LXG genes in human gut metagenomes

The 240 nucleotide tags from the toxin domains were mapped using blastn to the Integrated Gene Catalog (*Li et al., 2014*) – a large dataset of previously identified microbiome genes and their abundances in several extensive microbiome studies (including HMP [*Human Microbiome Project Consortium, 2012*], MetaHiT [*Qin et al., 2010*], and a T2D Chinese cohort [*Qin et al., 2012*]). Genes to which at least one tag was mapped with >95% identity and >50% overlap were labeled as LXG genes. This set of LXG genes was further manually curated to filter out genes that lack the LXG targeting domain. In analyzing the relative abundance of the LXG genes across samples, relative abundances $< 10^{-7}$ were assumed to represent noise and were set to 0. LXG genes that were not present above this threshold in any sample and samples with no LXG genes were excluded from the analysis.

## Determination of cellular NAD$^+$ levels

Measurement of cellular NAD$^+$ levels was performed as reported previously (*Whitney et al., 2015*). Briefly, *E. coli* strains harboring expression plasmids for Tse2, Tse6$_{tox}$, TelB$_{tox}$*, TelB$_{tox}$$^{R626A}$, TelB$_{tox}$*–TipB and a vector control were grown in LB media at 37°C to mid-log phase prior to induction of protein expression with 0.1% (*w/v*) rhamnose. 1 hr post-induction, cultures were diluted to $OD_{600}$ = 0.5 and 500 µL of cells were harvested by microcentrifugation. Cells were then lysed in 0.2 M NaOH, 1% (w/v) cetyltrimethylammonium bromide (CTAB) followed by treatment with 0.4 M HCl at 60°C for 15 min. After neutralization with 0.5 M Tris base, samples were then mixed with an equal volume of NAD/NADH-Glo Detection Reagent (Promega) prepared immediately before use as per the instructions of the manufacturer. Luciferin bioluminescence was measured continuously using a Synergy H1 plate reader. The slope of the luciferin signal from the linear range of the assay was used to determine relative NAD$^+$ concentration compared to a vector control strain.

## NADase assay

*S. intermedius* strains were grown to late-log phase before cells were removed by centrifugation at 3000 *g* for 15 min. Residual particulates were removed by vacuum filtration through a 0.2 um membrane and the resulting supernatants were concentrated 100-fold by spin filtration (30 kDa MWCO). NADase assays were carried out by mixing 50 µL of concentrated supernatant with 50 µL of PBS containing 2 mM NAD$^+$ followed by incubation at room temperature for 2 hr. Reactions were terminated by the addition of 50 µL of 6M NaOH and incubated in the dark at room temperature for 15 min. Samples were analyzed by UV light at a wavelength of 254 nm and imaged using a FluorChemQ (ProteinSimple). Relative NAD$^+$ consumption was determined using densitometry analysis of each of the indicated strain supernatants using the ImageJ software program (https://imagej.nih.gov/ij/).

## Bacterial toxicity experiments

To assess TelA and TelB toxicity in bacteria, stationary phase cultures of *E. coli* BL21 pLysS harboring the appropriate plasmids were diluted $10^6$ and each 10-fold dilution was spotted onto 3% LB agar plates containing the appropriate antibiotics. 0.1% (*w/v*) L-rhamnose and 100 µM IPTG were added to the media to induce expression of toxin and immunity genes, respectively. For TelB, plasmids containing the wild-type toxin domain (under non-inducing conditions) were not tolerated. To circumvent this, SOE pcr was used to assemble a variant (H661A) that was tolerated under non-induced conditions. Based on the similarity of TelB$_{tox}$ to *M. tuberculosis* TNT toxin, this mutation likely reduces the binding affinity of TelB to NAD$^+$ (*Sun et al., 2015*). To generate a TelB variant that exhibited significantly reduced toxicity under inducing conditions, a second mutation (R626A) was introduced in the toxin domain of TelB. For examination of TelC toxicity in *S. intermedius*, the gene fragment encoding TelC$_{tox}$ was fused to the constitutive P96 promoter followed by a start codon and cloned into pDL277 (*Lo Sapio et al., 2012*). For extracellular targeting of TelC$_{tox}$ in *S.*

*intermedius*, the gene fragment encoding the sec-secretion signal (residues 1–30) of *S. pneumoniae* LysM (SP_0107) was fused to the 5' end of *telC_{tox}*, each of the *telC_{tox}* site-specific variants and the *telC_{tox}*–*tipC* bicistron. 500 ng of each plasmid was transformed in *S. intermedius* B196 and toxicity was assessed by counting the number of transformants. For examination of TelC toxicity *S. aureus*, the gene fragment encoding TelC_{tox} was cloned into the xylose-inducible expression vector pEPSA5. For extracellular targeting, the gene fragment encoding the sec-secretion signal for *hla* was fused to the 5' end of *telC_{tox}* and *telC_{tox}^{D401A}*. TelC-based toxicity was assessed in the same manner as was done for the above *E. coli* toxicity experiments except that xylose (2% *w/v*) was included in the media to induce protein expression. Detailed plasmid information can be found in *Supplementary file 2*.

## Time-lapse microscopy

*S. aureus* USA300 pEPSA5::*ss-telC_{202-CT}* and *S. aureus* USA300 pEPSA5 were resuspended in TSB and 1–2 µL of each suspension was spotted onto an 1% (*w/v*) agarose pad containing typtic soy medium supplemented with 2% (*w/v*) xylose and sealed.

Microscopy data were acquired using NIS Elements (Nikon) acquisition software on a Nikon Ti-E inverted microscope with a 60× oil objective, automated focusing (Perfect Focus System, Nikon), a xenon light source (Sutter Instruments), and a CCD camera (Clara series, Andor). Time-lapse sequences were acquired at 10 min intervals over 12 hr at room temperature. Movie files included are representative of three biological replicates for each experiment.

## Extracellular proteome

200 mL cultures of *S. intermedius* B196 wild-type and *ΔessC* strains were grown to stationary phase in THYB before being pelleted by centrifugation at 2500 × *g* for 20 min at 4°C. Supernatant fractions containing secreted proteins were collected and spun at 2500 × *g* for an additional 20 min at 4°C and subsequently filtered through a 0.2 µm pore size membrane to remove residual cells and cell debris. Protease inhibitors (1 mM AEBSF, 10 mM leupeptin, and 1 mM pepstatin) were added to the filtered supernatants prior to dialysis in 4L of PBS using 10 kDa molecular weight cut off tubing at 4°C. After four dialysis buffer changes, the retained proteins were TCA precipitated, pelleted, washed in acetone, dried and resuspended in 1 mL of 100 mM ammonium bicarbonate containing 8 M urea. The denatured protein mixture was then desalted over a PD10 column prior to reduction, alkylation and trypsin digestion as described previously (*Eshraghi et al., 2016*). The resulting tryptic peptides were desalted and purified using C18 spin columns (Pierce) following the protocol of the manufacturer before being vacuum dried and resuspended in 10 µL of acetonitrile/$H_2O$/formic acid (5/94.9/0.1, *v/v/v*) for LC-MS/MS analysis.

Peptides were analyzed by LC-MS/MS using a Dionex UltiMate 3000 Rapid Separation nanoLC and a linear ion trap – Orbitrap hybrid mass spectrometer (ThermoFisher Scientific). Peptide samples were loaded onto the trap column, which was 150 µm x 3 cm in-house packed with 3 µm C18 beads, at flow rate of 5 µL/min for 5 min using a loading buffer of acetonitrile/$H_2O$/formic acid (5/94.9/0.1, *v/v/v*). The analytical column was a 75 µm x 10.5 cm PicoChip column packed with 1.9 µm C18 beads (New Objectives). The flow rate was kept at 300 nL/min. Solvent A was 0.1% formic acid in water and Solvent B was 0.1% formic acid in acetonitrile. The peptide was separated on a 90 min analytical gradient from 5% acetonitrile/0.1% formic acid to 40% acetonitrile/0.1% formic acid.

The mass spectrometer was operated in data-dependent mode. The source voltage was 2.10 kV and the capillary temperature was 275°C. $MS^1$ scans were acquired from 400 to 2000 m/z at 60,000 resolving power and automatic gain control (AGC) set to $1 \times 10^6$. The top ten most abundant precursor ions in each $MS^1$ scan were selected for fragmentation. Precursors were selected with an isolation width of 1 Da and fragmented by collision-induced dissociation (CID) at 35% normalized collision energy in the ion trap. Previously selected ions were dynamically excluded from re-selection for 60 s. The $MS^2$ AGC was set to $3 \times 10^5$.

Proteins were identified from the MS raw files using Mascot search engine (Matrix Science). MS/MS spectra were searched against the UniprotKB database of *S. intermedius* B196 (*UniProt and UniProt Consortium, 2015*). All searches included carbamidomethyl cysteine as a fixed modification and oxidized Met, deamidated Asn and Gln, acetylated N-terminus as variable modifications. Three missed tryptic cleavages were allowed. The $MS^1$ precursor mass tolerance was set to 10 ppm and

removed by centrifugation at 30,000 $g$ for 45 min. Cleared cell lysates were then purified by nickel affinity chromatography using 2 mL of Ni-NTA agarose resin loaded onto a gravity flow column. Lysate was loaded onto the column and unbound proteins were removed using 50 mL of buffer A. Bound proteins were then eluted using 50 mM Tris-HCl pH 8.0, 300 mM NaCl, 400 mM imidazole. The purity of each protein sample was assessed by SDS-PAGE followed by Coomassie Brilliant Blue staining. All protein samples were dialyzed into 20 mM Tris-HCl, 150 mM NaCl.

Selenomethionine-incorporated TelC$_{202\text{-CT}}$ was obtained by growing *E. coli* BL21 pETDuet-1:: *telC*$_{202\text{-CT}}$ in SelenoMethionine Medium Complete (Molecular Dimensions) using the expression conditions described above. Cell lysis and nickel affinity purification were also performed as described above except that all buffers contained 1 mM tris(2-carboxyethyl)phosphine.

## Crystallization and structure determination

Purified selenomethionine-incorporated TelC$_{202\text{-CT}}$ was concentrated to 12 mg/mL by spin filtration (10 kDa cutoff, Millipore) and screened against commercially available crystallization screens (MCSG screens 1–4, Microlytic). Diffraction quality crystals appeared after 4 days in a solution containing 0.1 M Sodium Acetate pH 4.6, 0.1 M CaCl$_2$, 30% PEG400. X-ray diffraction data were collected using beamline 5.0.2 at the Advanced Light Source (ALS). A single dataset (720 images, 1.0° $\Delta\varphi$ oscillation, 1.0 s exposure) was collected on an ADSC Q315r CCD detector with a 200 mm crystal-to-detector distance. Data were indexed and integrated using XDS (*Kabsch, 2010*) and scaled using AIMLESS (*Evans and Murshudov, 2013*) (table S2).

The structure of TelC$_{202\text{-CT}}$ was solved by Se-SAD using the AutoSol wizard in the Phenix GUI (*Adams et al., 2010*). Model building was performed using the AutoBuild wizard in the Phenix GUI. The electron density allowed for near-complete building of the model except for N-terminal residues 202–211, two C-terminal residues and an internal segment spanning residues 417–434. Minor model adjustments were made manually in COOT between iterative rounds of refinement, which was carried out using Phenix.refine (*Afonine et al., 2012*; *Emsley et al., 2010*). The progress of the refinement was monitored by the reduction of $R_{\text{work}}$ and $R_{\text{free}}$ (*Table 2*).

## Peptidoglycan hydrolase assay

Purified TelC$_{\text{tox}}$ was dialyzed against 20 mM sodium acetate pH 4.6, 150 mM NaCl, 10 mM CaCl$_2$. Cross-linked peptidoglycan sacculi and lysostaphin endopeptidase pre-treated (non-cross-linked) sacculi from *S. aureus* were then incubated with 5 µM TelC$_{\text{tox}}$, 2.5 µg of cellosyl muramidase or buffer at 37°C for 18 hr. Digests were then boiled for 5 min at 100°C and precipitated protein was removed by centrifugation. The resulting muropeptides were reduced by the addition of sodium borohydride and analyzed by HPLC as described previously (*de Jonge et al., 1992*).

For the analysis of cell walls isolated from TelC-intoxicated cells, 1L of *S. aureus* USA300 pEPSA5::*ss-telC*$_{\text{tox}}$ and *S. aureus* USA300 pEPSA5::*ss-telC*$_{\text{tox}}^{\text{D401A}}$ cells were grown to mid-log phase prior to induction of protein expression by the addition of 2% (*w/v*) xylose. 90 min post-induction, cultures were rapidly cooled in an ice-water bath and cells were harvested by centrifugation. After removal of supernatants, cell pellets were resuspended in 40 mL of ice-cold 50 mM Tris-HCl pH 7.0 and subsequently added dropwise to 120 mL boiling solutions of 5% SDS. PG was isolated as described (*de Jonge et al., 1992*) and digested with either cellosyl muramidase or lysostaphin endopeptidase and cellosyl, reduced with sodium borohydride and analyzed by HPLC as described above.

## Lipid II phosphatase assay

Purified TelC$_{\text{tox}}$ and TelC$_{\text{tox}}$–TipC$_{\Delta\text{ss}}$ complex were dialyzed against 20 mM sodium acetate pH 4.6, 150 mM NaCl, 10 mM CaCl$_2$. C$^{14}$-labelled Lys-type lipid II was solubilized in 5 µL of Triton X-100 before being added to 95 µL of reaction buffer containing 15 mM HEPES pH 7.5, 0.4 mM CaCl$_2$ (excluded from the PBP1B-LpoB reaction), 150 mM NaCl, 0.023% Triton X-100 and either PBP1B– LpoB complex, TelC$_{\text{tox}}$, TelC$_{\text{tox}}$–TipC$_{\Delta\text{ss}}$ complex or Colicin M followed by incubation for 1 hr at 37°C. The reaction with PBP1B-LpoB was boiled and reduced with sodium borohydride. All the reactions were quenched by the addition of 1% (*v/v*) phosphoric acid and analyzed by HPLC as described (*Bertsche et al., 2005*). Three biological replicates were performed for each reaction. The lipid II

degradation products of TelC$_{tox}$ digestion were confirmed by mass spectrometry. Lipid II was kindly provided by Ute Bertsche and was generated as described previously (*Bertsche et al., 2005*).

For thin-layer chromatography (TLC) analysis of Lys-type lipid II degradation by TelC, TelC$_{tox}$ or TelC$_{tox}$–TipC$_{\Delta ss}$, 40 µM lipid II was solubilized in 30 mM HEPES/KOH pH 7.5, 150 mM KCl and 0.1% Triton X-100 before adding either 2 µM TelC$_{tox}$, 2 µM TelC$_{tox}$–TipC$_{\Delta ss}$ complex or protein buffer, followed by incubation for 90 min at 37°C. Samples were extracted with n-butanol/pyridine acetate (2:1) pH 4.2 and resolved on silica gel (HPTLC silica gel 60, Millipore) in chloroform/methanol/ammonia/water (88:48:1:10). For the undecaprenyl phosphate reactions 100 µM undecaprenyl phosphate (Larodan) was solubilized in 20 mM HEPES/KOH pH 7.5, 150 mM KCl, 1 mM CaCl$_2$ and 0.1% Triton X-100 before adding 2 µM TelC$_{tox}$ (final concentration), 2 µM TelC$_{tox}$–TipC$_{\Delta ss}$ or protein buffer, followed by incubation for 5 hr at 25°C and 90 min at 37°C. Samples were extracted and separated by TLC as indicated above. For the undecaprenyl pyrophosphate synthesis reactions coupled to the degradation by TelC$_{tox}$0.04 mM Farnesyl pyrophosphate and 0.4 mM isopentenyl pyrophosphate were solubilized in 20 mM HEPES/KOH pH 7.5, 50 mM KCl, 0.5 mM MgCl$_2$, 1 mM CaCl$_2$, 0.1% Triton X-100 and incubated with 10 µM UppS and 2 µM TelC$_{tox}$ (final concentrations) or protein buffer for 5 hr at 25°C and 90 min at 37°C. Samples were extracted and separated by TLC as indicated above.

## Yeast toxicity assay

To target TelC to the yeast secretory pathway, *telC$_{tox}$* was fused to the gene fragment encoding the leader peptide of *Kluyveromyces lactis* killer toxin (*Baldari et al., 1987*), generating *ss-telC$_{tox}$*. *S. cerevisiae* was transformed with pCM190 containing *telC$_{tox}$*, *ss-telC$_{tox}$*, a known toxin of yeast or empty vector and grown o/n SC-ura +1 ug/mL doxycycline. Cultures were resuspended to OD$_{600}$ = 1.5 with water and serially diluted 5-fold onto SC-ura agar. Plates were incubated at 30°C for 2 days before being imaged using a Pentax WG-3 digital camera. Images presented are representative of three independent replicate experiments. Proteolytic processing of the leader peptide of ss-TelC$_{tox}$ was confirmed by western blot.

## Isothermal titration calorimetry

Solutions of 25 µM TelC$_{202\text{-}CT}$ and 250 µM TipC$_{\Delta ss}$ were degassed prior to experimentation. ITC measurements were performed with a VP-ITC microcalorimeter (MicroCal Inc., Northampton, MA). Titrations consisted of 25 10 µL injections with 180 s intervals between each injection. The ITC data were analyzed using the Origin software package (version 5.0, MicroCal, Inc.) and fit using a single-site binding model.

## Bacterial two-hybrid analyses

*E. coli* BTH101 cells were co-transformed with plasmids encoding the T18 and T25 fragments of *Bordetella pertussis* adenylate cyclase fused to the proteins of interest. Stationary phase cells were then plated on LB agar containing 40 mg/mL X-gal, 0.5 mM IPTG, 50 mg/mL kanamycin and 150 mg/mL carbenicillin and grown for 24 hr at 30°C. Plates were imaged using a Pentax WG-3 digital camera. Images representative of at least three independent replicate experiments are presented.

## Immunoprecipitation assay

*E. coli* BL21 (DE3) pLysS cells were co-transformed with plasmids encoding TelC-his$_6$ and WxgB-V or TelC-his$_6$ and WxgC-V. Cells were grown to an OD$_{600}$ of 0.6 prior to induction of protein expression with 0.5 mM IPTG for 6 hr at 30°C. Cultures were harvested by centrifugation and cell pellets were resuspended in Buffer A prior to lysis by sonication. Clarified lysates were then incubated with Ni-NTA resin and incubated at 4°C with rotation for 90 min. Ni-NTA resin was then washed four times with Buffer A followed by elution of bound proteins with Buffer B. After the addition of Laemmli sample buffer, proteins were separated by SDS-PAGE using an 8–16% gradient TGX Stain-Free gel (Bio-Rad). TelC-his$_6$ was visualized by UV activation the trihalo compound present in Stain-Free gels whereas WxgB-V and WxgC-V were detected by western blotting.

## Acknowledgements

We thank Eefjan Breukink for providing lipid II, Joe Gray for mass spectrometry analyses, the Fang and Rajagopal laboratories for sharing reagents, Lynn Hancock and Gary Dunny for providing strains and plasmids, Corie Ralston for help with X-ray data collection, Ben Ross and Simon Dove for critical reading of the manuscript, and members of the Mougous laboratory for helpful discussions. This work was funded by the NIH (AI080609 to JDM and AT007802 to EB), the Defense Threat Reduction Agency (HDTRA1-13-1-0014 to JDM), and the Medical Research Council (MR/N0026791/1). Proteomics services were performed by the Northwestern Proteomics Core Facility, generously supported by NCI CCSG P30 CA060553 awarded to the Robert H Lurie Comprehensive Cancer Center and the National Resource for Translational and Developmental Proteomics supported by P41 GM108569. JCW was supported by a postdoctoral research fellowship from the Canadian Institutes of Health Research, AJV was supported by a postdoctoral fellowship from the Natural Science and Engineering Research Council of Canada, and JDM holds an Investigator in the Pathogenesis of Infectious Disease Award from the Burroughs Wellcome Fund and is an HHMI Investigator.

## Additional information

### Funding

| Funder | Grant reference number | Author |
| --- | --- | --- |
| Canadian Institutes of Health Research | | John C Whitney |
| Natural Sciences and Engineering Research Council of Canada | | Adrian J Verster |
| National Cancer Institute | CCSG P30 CA060553 | Young Ah Goo |
| National Institutes of Health | AT007802-01 | Elhanan Borenstein |
| Medical Research Council | MR/N0026791/1 | Waldemar Vollmer |
| National Institutes of Health | AI080609 | Joseph D Mougous |
| Howard Hughes Medical Institute | | Joseph D Mougous |
| Burroughs Wellcome Fund | 1010010 | Joseph D Mougous |
| Defense Threat Reduction Agency | HDTRA1-13-1-0014 | Joseph D Mougous |

The funders had no role in study design, data collection and interpretation, or the decision to submit the work for publication.

### Author contributions

JCW, Conceptualization, Investigation, Writing—original draft, Writing—review and editing; SBP, Conceptualization, Supervision, Funding acquisition, Investigation, Writing—original draft, Writing—review and editing; JK, MP, AJV, MCR, YAG, Investigation, Writing—review and editing; HDK, MQC, NPB, DB, MGS, Investigation; EB, WV, Funding acquisition, Investigation, Writing—review and editing; JDM, Conceptualization, Supervision, Funding acquisition, Investigation, Writing—original draft, Project administration, Writing—review and editing

### Author ORCIDs

John C Whitney, http://orcid.org/0000-0002-4517-8836
S Brook Peterson, http://orcid.org/0000-0003-2648-0965
Jungyun Kim, http://orcid.org/0000-0003-3793-4264
Joseph D Mougous, http://orcid.org/0000-0002-5417-4861

## Additional files

**Supplementary files**
• Supplementary file 1. Strains used in this study.
• Supplementary file 2. Plasmids used in this study.

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
