## [Decision Letter]

Thank you for submitting your article "A broadly distributed toxin family mediates contact-dependent antagonism between gram-positive bacteria" for consideration by *eLife*. Your article has been reviewed by three peer reviewers, one of whom is a member of our Board of Reviewing Editors, and the evaluation has been overseen by Gisela Storz as the Senior Editor. The following individual involved in review of your submission has agreed to reveal his identity: Jeffery S Cox (Reviewer #3).

The reviewers have discussed the reviews with one another and the Reviewing Editor has drafted this decision to help you prepare a revised submission.

Summary:

In this paper, the authors investigate the LXG protein family to test the overarching hypothesis that LXG proteins are toxins used for inter-species bacterial antagonism. Specifically, they focus on two LXG proteins produced by *Streptococcus intermedius* (TelB and TelC). They show that TelB and TelC are independently secreted by the Esx pathway in *S. intermedius*. The authors claim that TelB has NADase activity, and demonstrate that TelC is a calcium phosphatase acting on the bacterial peptidoglycan precursor Lipid II. Their data show that Esx-mediated secretion of TelC and TelB is likely contact-dependent, and that TelB mediates inter-species bacterial antagonism. The authors also present intriguing but preliminary data that WXG100-like proteins interact with their cognate LXG domain proteins, and that WXG100-like proteins promote LXG toxin secretion. Although the reviewers were enthusiastic about some aspects of the work, a number of shortcomings were identified that would need to be addressed if the authors wish to resubmit.

Essential revisions:

1) Although the paper was submitted as a 'Short Report', this length restriction led to some results being poorly explained or, in some cases like the WXG-100 data, almost neglected. There was also an extremely limited introduction, which is not ideal for a broad interest journal like *eLife*. The authors should flesh out the Introduction, Results, and Discussion to make the paper more accessible to a wider range of readers.

2) There was some confusion about a fundamental aspect of this system: On the one hand, the authors have argued that LXG proteins can only inhibit via a contact-dependent mechanism, but the proteins are easily detected in the supernatant suggesting they're constitutively secreted. This seems wasteful – why would a contact-dependent inhibition system spew toxin into the environment when it can only be active via direct delivery? Maybe more importantly, can the authors rule out the possibility that toxins are secreted and can kill in a contact-independent way, but are somehow getting stuck on the membranes used for the filter-well experiments? On a related note, the authors claim at one point that "The discovery that LXG toxin family members are delivered between bacteria…" But there's no direct evidence that TelC is transferred to recipient cells. The co-culturing experiments are consistent with such a model, but not definitive. For instance, are the fitness effects observed in co-culture on solid media really due to "intoxication" of the recipient strain? Is it possible to over-express the antitoxins in the recipient cell, in the experiments presented in both Figure 2? Likewise, it would be nice to show specificity of the two systems (TelB/TipB and TelC/TipC) by performing competitive experiments with all combinations of mutants. In short, stronger, more direct evidence is needed that the LXG toxins are transferred from donors to recipients.

3) It would appear that Video 2 only shows what happens when ToxCtox is expressed by cells, i.e. self-intoxication after overexpression. But ToxC is normally is secreted into the extracellular medium. So the authors need to show that it can be toxic 'in trans', presumably by co-culturing cells on a solid surface given that co-culturing in liquid doesn't work. This is important to concluding that, if/when transferred as a full-length protein (not just the tox domain used in Video 2), ToxC targets lipid II.

4) The authors tested to see if production of the toxin TelC, which cleaves lipid II, is toxic in yeast cells. They were curious about this because these eukaryotes produce essential polyisoprenoid-linked glycoconjugates that are required for N-glycosylation. The authors do not see any toxicity. It is possible that TelC is not toxic because it cannot hydrolyze those essential lipid-linked saccharides. However, there are a lot of trivial reasons that could explain the lack of toxicity. For example, most of the protein might not be properly targeted since most of it still has a signal sequence (Figure 3). A better test to determine substrate specificity would be to determine if TelC can cleave other bacterial undecaprenyl-linked saccharides. Although ideal, such work is not necessary and if not done, the authors should remove the text and Figure 3 panels G and H corresponding to the aforementioned test.

5) The somewhat grander claims regarding the implied role these LXG proteins play in polymicrobial communities is perhaps a bit over-speculative. In general, it seem that there is a disconnect between the data the authors present and their emphasis in the Abstract (and throughout the manuscript) on human gut metagenomes and polymicrobial dynamics. Throughout the manuscript, the authors emphasize the abundance/prevalence of LXG proteins in human microbiomes, but those data are neither very convincing nor novel. Thus, the authors have overstated the implications of the inter-species antagonism mediated by Esx secretion, as they do not present data illustrating the functional outcome of these interactions in an actual polymicrobial community.

6) The authors conclude that LXG toxins are specific to Gram + inter-species antagonism, but have only tested a single Gram negative species (*E. coli*). Testing a broader range of Gram negatives would be an easy way to bolster this claim.

7) The notion that TelB is a NAD+ degrading toxin is less than compelling. Two important controls are missing in Figure 2 – mutation of the conserved residues that comprise the active site as predicted in Figure 2—figure supplement 2, and rescue of the toxic effect by co-expressing TipB. Complementation of the Si telB mutant using wild-type and mutant forms of TelB would also be important in this regard as well. Finally, the assay in Figure 2 is somewhat unconvincing. An in vitro NADase activity assay with purified TelB would be much more convincing and quantitative.

8) Given that the main emphasis in the paper is on TelC, it is disappointing that the authors do not demonstrate inter-species antagonism dependent on TelC. Perhaps we missed something, but it appears that the data in Figure 2 are all experiments done with S.i. strains. In Figure 2, they show that a strain WT for TelB elicits the crucial inter-species antagonism, but it is not clear whether this strain is also making TelC?

9) The idea that all three LXG proteins are secreted from Si via Esx secretion is not well supported. Table 1 is confusing – the "locus tag" is not intuitive, making it difficult to understand which proteins are represented in the MS results. Importantly, it is critical to show that secreted proteins that are not associated with Esx are present in similar levels in wild-type and essC mutant supernatants.

10) Figure 4 is compelling but underdeveloped. The exclusive use of bacterial two-hybrid seems insufficient to show convincingly that the LSG and WXG proteins interact specifically, and thus any conclusions at this point seems speculative.

11) In Figure 2, why does the CI decrease to well below 1 when the filter is present?

12) Roughly how much TelC is secreted in an exponential or overnight culture? It is not clear how to interpret Figure 2—figure supplement 3 where bacteria are incubated with purified "active toxin" without an estimate of the amount of toxin produced endogenously. Also, the authors don't show data to indicate that their purified recombinant TelC protein is active.

13) In Figure 1, the authors highlight another Streptococcus species (parasanguinus) that has an LXG protein which they annotate as a Lipid II phosphatase. That raises the question of how novel their finding about TelC in Si is? This deserves clarification.

[Editors' note: further revisions were requested prior to acceptance, as described below.]

Thank you for resubmitting your work entitled "A broadly distributed toxin family mediates contact-dependent antagonism between gram-positive bacteria" for further consideration at *eLife*. Your revised article has been favorably evaluated by Gisela Storz as Senior editor, a Reviewing editor, and two reviewers.

The manuscript has been improved and the consensus is that the paper is suitable for *eLife* provided you can address the additional issues raised by one of the reviewers:

*Reviewer #2:*

The manuscript has improved significantly because of the re-write and the inclusion of new data. I appreciate the authors' detailed responses to the reviews and their efforts to address the issues. Overall, I think they addressed the main concerns satisfactorily. I would only like to request a few modifications to the text.

The main comment I have is about the cellular target of TelC. The authors show that TelC: a) needs to be exported to be toxic in both *E. coli* and Staph, and b) degrades both lipid II and undecaprenyl-PP (C55-PP) in vitro. They conclude that, in cells, TelC targets lipid II that is located in the outer leaflet of the cytoplasmic membrane. This could lead readers to mistakenly think that lipid II is only present at that location and that C55-PP is not. In reality, C55-PP is also present in the outer leaflet of the cytoplasmic membrane, and both lipid II and C55-PP are present in both leaflets of the cytoplasmic membrane. Therefore, I suggest that the authors state that their in vivo data cannot distinguish whether TelC targets lipid II and/or undecaprenyl-PP (C55-PP) and that it is unknown why toxicity is limited to an extracytoplasmic location given that these essential peptidoglycan intermediates transit through both leaflets of the cytoplasmic membrane. [A simple explanation for their data could be that the intermediate(s) is not accessible in the cytoplasmic side of the membrane because proteins that normally handle it during peptidoglycan synthesis protect it from TelC or the intermediate has a very short residence time at that location].

---

## [Author Response]

*Essential revisions:*

*1) Although the paper was submitted as a 'Short Report', this length restriction led to some results being poorly explained or, in some cases like the WXG-100 data, almost neglected. There was also an extremely limited introduction, which is not ideal for a broad interest journal like eLife. The authors should flesh out the Introduction, Results, and Discussion to make the paper more accessible to a wider range of readers.*

We agree wholeheartedly and have performed a complete overhaul of the text. Among other improvements described below, at the suggestion of the reviewers, we have extended the Introduction, Results, and Discussion sections of the revised manuscript. We regret the lack of clarity brought about by the brevity of our original submission.

*2) There was some confusion about a fundamental aspect of this system: On the one hand, the authors have argued that LXG proteins can only inhibit via a contact-dependent mechanism, but the proteins are easily detected in the supernatant suggesting they're constitutively secreted. This seems wasteful – why would a contact-dependent inhibition system spew toxin into the environment when it can only be active via direct delivery? Maybe more importantly, can the authors rule out the possibility that toxins are secreted and can kill in a contact-independent way, but are somehow getting stuck on the membranes used for the filter-well experiments? On a related note, the authors claim at one point that "The discovery that LXG toxin family members are delivered between bacteria…" But there's no direct evidence that TelC is transferred to recipient cells. The co-culturing experiments are consistent with such a model, but not definitive. For instance, are the fitness effects observed in co-culture on solid media really due to "intoxication" of the recipient strain? Is it possible to over-express the antitoxins in the recipient cell, in the experiments presented in both Figure 2? Likewise, it would be nice to show specificity of the two systems (TelB/TipB and TelC/TipC) by performing competitive experiments with all combinations of mutants. In short, stronger, more direct evidence is needed that the LXG toxins are transferred from donors to recipients.*

The reviewers pose an interesting question, but one that is difficulty to answer. Prior to speculating why a bacterium would secrete toxins that require cell-cell contact for delivery into the milieu during propagation in vivo, it is worth clarifying a few points. First, by spectral counts deriving from mass spectrometry data (Table 1), the abundance of the LXG proteins is below the average for other proteins we detect in culture supernatants (TelA-C, μ=12.3; All others, μ=26.50). Whereas proteins that are abundantly secreted by bacteria can be detected in unconcentrated culture supernatants using SDS-PAGE followed by coomassie stain (e.g. see Jacob-Dubuisson et al., Microbiol. 2000), we detect TelC by Western blot only after TCA precipitation of culture supernatants (e.g. see Figure 2 and Figure 5). Second, it is worth noting that the detection of LXG proteins in the supernatant may not be an indication that they are constitutively secreted. The production and/or export of the protein may still be elevated or repressed considerably depending on conditions.

Though counterintuitive, there exists overwhelming precedent for cell contact-dependent secretion systems that release their substrates to detectable levels in the absence of recipient cells. This has been documented for (note – not an exhaustive list): type III secretion (Nikolaus et al., J. Bacteriol. 2001/Rietsch et al., PNAS, 2005/Anderson and Schneewind, Mol. Microbiol.,1999) type IV secretion (Souza et al., Nat. Commun. 2015) type VI secretion (English et al., Mol. Microbiol. 2012/Eshraghi et al., 2016/Schwarz et al. Infect. Immun. 2014)

Indeed, our own group has used this property of the type VI secretion systems of *Francisella novicida* and *Burkholderia thailandensis* to define the effector proteins released by these systems (Eshraghi et al., Cell Host Microbe. 2016/Schwarz et al. Infect. Immun. 2014). In some cases, a specific cue (e.g. calcium removal, potassium addition) is required for activation of the system. Since we do not know if such a cue affects the Esx pathway of *S. intermedius*, we cannot determine if we have fortuitously overridden strict negative regulation by inadvertently providing an activation cue in the in vitro growth conditions we utilize.

Despite the considerations mentioned above, we have contemplated evolutionary mechanisms that could provide an explanation for apparent constitutive export of proteins that are dependent on cell contact for efficacy. We speculate that bacteria that routinely face contacting competitor cells may evolve to relieve repression of cell contact-mediated toxin delivery systems. There is evidence for this in the type VI secretion systems of *Pseudomonas aeruginosa* and *Vibrio cholera.* Although many strains of these organisms display strongly repressed type VI secretion systems when grown in vitro, other isolates possess highly active systems under the same conditions (*P. aeruginosa* clinical isolates studied in Mougous et al., Nat. Cell Biol. and the V52 strain of *V. cholerae* identified in Pukatzki et al.,Proc. Natl. Acad. Sci. 2006). This may be the case for the *S. intermedius* strain we employ in the current study. Anecdotally, we can share that prior to working with *S. intermedius*, we failed to observe Esx-dependent LXG protein export in several organisms. We suspect that many of the challenges we encountered are a result of secretion system inactivity under the conditions we utilized.

Respectfully, we believe that within reason we have ruled out the possibility that secreted TelC kills in the absence of a functional Esx system and cell contact. In addition to the membrane experiment referred to by the reviewer, our original submission included the demonstration that purified, full-length TelC, added to a final concentration of 0.1 mg/mL has no impact on the growth of TelC-susceptible cells (Figure 3—figure supplement 2). We also showed that highly concentrated TelB and TelC donor cells (OD_600nm_ = 20) are unable to inhibit the growth of susceptible recipient cells in liquid media (Figure 3—figure supplement 2). Finally, as noted, we showed that a 0.2 μm pore diameter membrane separating donor and sensitive recipient cells abrogates the Esx-dependent growth advantage (Figure 3).

We cannot rule out that LXG proteins are adhering to the 0.2 μm pore diameter membrane during our competition experiments. However, these are low protein binding track-etched polycarbonate membranes and we would expect that the minimal protein binding capacity they possess would be rapidly saturated by virtue of being coated with cells, rich media, and a variety of extracellular proteins. Nevertheless, to address this potential confounding factor, we experimentally measured the propensity of TelC to adsorb to the membranes used in our contact-dependency studies (Figure 6). In summary, we failed to detect TelC adsorption to these membranes.

Author response image 1.TelC does not irreversibly associate with polycarbonate membranes to a detectable level.The indicated quantity of TelC (Before) was incubated with 16 cm^2^ of filter material in PBS. After an extended incubation, the filter was removed and the protein was again quantified (After).**DOI:**
http://dx.doi.org/10.7554/eLife.26938.022

We agree that we have not provided direct evidence of LXG transport into recipient cells. However, we believe that no other reasonably plausible mechanism is consistent with the totality of our data and we now strengthen this further with new data described in the subsequent paragraph. We show conclusively that two LXG proteins are toxins exported via the Esx secretion system (Figure 2), and that the export of these proteins is required for their capacity to inhibit the proliferation of recipient cell populations (Figure 3). We also show that this growth inhibitory activity of the toxins occurs only in strain populations rendered susceptible to the toxins by the removal of genes that encode proteins that bind to (in the case of TelC–TipC this is demonstrated in our paper in Figure 3—figure supplement 1) and inhibit the toxin-associated biochemical activities (both TelC–TipC and TelB–TipB are demonstrated in this study, see Figure 3).

To add to the data in our original manuscript, we now provide evidence of toxin-mediated cell lysis occurring in co-cultures between wild-type *Si* and a mutant rendered susceptible to TelC (Figure 7,described in detail in Reviewer Response to Essential Revision 3). We have additionally ruled out the possibility that secondary site mutations cause general growth defects in the *telB tipB* and *telC tipC* mutant strains by performing whole genome sequencing (new whole genome sequencing data described in the legend to Figure 3), and by demonstrating these strains compete equally with the wild-type parent in liquid culture (Figure 3). Additionally, we find that the difference in competitive index between a wild-type strain and an *essC*-inactivated strain towards a susceptible recipient species can be attributed entirely to changes in the final growth yield of the recipient, consistent with growth inhibition mediated by toxin delivery (Figure 8). Finally, it is worth noting that there is precedent for similar experiments as those we present here being used to first demonstrate contact-dependent intercellular toxin delivery by a specialized secretion systems (Souza et al., Nat. Comm. 2015 / Hood et al., Cell Host Microbe 2010 / Aoki et al. Science. 2005).

Author response image 2.Frequency of double-resistant colonies resulting from co-culture of a streptomycin resistant donor and a kanamycin resistant recipient is elevated under conditions conducive to intoxication via intracellular transfer of TelC.**DOI:**
http://dx.doi.org/10.7554/eLife.26938.023

Author response image 3.The Esx-pathway of *S. intermedius* mediates inhibition of *E. faecalis* growth.CFUs were determined by plating on selective media after 24 hours growth of co-cultures. Populations of *E. faecalis* were equal between cultures at the start of the experiment.**DOI:**
http://dx.doi.org/10.7554/eLife.26938.024

*3) It would appear that Video 2 only shows what happens when ToxCtox is expressed by cells, i.e. self-intoxication after overexpression. But ToxC is normally is secreted into the extracellular medium. So the authors need to show that it can be toxic 'in trans', presumably by co-culturing cells on a solid surface given that co-culturing in liquid doesn't work. This is important to concluding that, if/when transferred as a full-length protein (not just the tox domain used in Video 2), ToxC targets lipid II.*

We assume that here and elsewhere, the reviewer is referring to TelC, as we did not study a protein designated ToxC. That aside, we believe there may be some confusion here, perhaps related to our oversight in not including legends for these videos. The bacteria in the video are *S. aureus*, and so we are not reporting native self-intoxication, but instead these cells are simply convenient Gram-positive heterologous hosts for observing potential phenotypic consequences of TelC_tox_ over-expression. Video 2 is presented to illustrate that the toxin domain of TelC can promote cellular lysis when over-expressed and artificially targeted to the exterior of the cell. This observation, in conjunction with our finding that the overall structure of the peptidoglycan sacculus remains largely intact, is how we came upon the hypothesis that TelC acts upstream in cell wall synthesis to degrade lipid II. In the revised manuscript, we provide biochemical evidence that full length TelC is active on lipid II (Figure 4—figure supplement 3). Nonetheless, we have now also obtained evidence that TelC-dependent cell lysis occurs during co-culture between *Si* bearing a streptomycin resistant cassette inserted at a neutral chromosomal location, and a mutant rendered susceptible to TelC through insertion of a kanamycin cassette in place of the *telC tipC* locus (Figure 7). We find that after 30 hours of co-culture of these strains, a significant proportion of the resulting population acquires resistance to both antibiotics. The frequency of double-resistant colonies obtained from co-cultures in which the streptomycin-resistant strain contained inactivated *telC* is significantly lower. It is well established in Streptococcus that cell lysis provides DNA for uptake by natural transformation. Thus, the accumulation of cells containing both markers is a useful proxy for cell lysis. These results provide further evidence that TelC is a bacteriolytic toxin.

*4) The authors tested to see if production of the toxin TelC, which cleaves lipid II, is toxic in yeast cells. They were curious about this because these eukaryotes produce essential polyisoprenoid-linked glycoconjugates that are required for N-glycosylation. The authors do not see any toxicity. It is possible that TelC is not toxic because it cannot hydrolyze those essential lipid-linked saccharides. However, there are a lot of trivial reasons that could explain the lack of toxicity. For example, most of the protein might not be properly targeted since most of it still has a signal sequence (Figure 3). A better test to determine substrate specificity would be to determine if TelC can cleave other bacterial undecaprenyl-linked saccharides. Although ideal, such work is not necessary and if not done, the authors should remove the text and Figure 3 panels G and H corresponding to the aforementioned test.*

We agree that there are trivial reasons for the apparent inability of TelC to inhibit yeast growth. However, we do believe that these negative data have value, as they show that contrary to the situation with bacteria, TelC is not toxic to a eukaryotic cell when expressed from a strong promoter. We show the toxin is produced to detectable levels in the cytoplasm and we provide evidence that our construct directing it to the secretory pathway is performing as expected. If this were a bona fide eukaryotic toxin, we would anticipate that levels detected by direct Western blot of lysed cells (immunoprecipitation not required) would likely be sufficient to induce detectable killing or growth inhibition, which we do not observe. This is an experiment that we feel others in the community would want to see – caveats notwithstanding – given that we are proposing TelC targets bacteria physiologically. We do appreciate the point raised by the reviewer, and so to meet in the middle on this matter, we have moved the yeast data to the supplement and we have simplified our interpretation (Figure 4—figure supplement 3). We now avoid the dolichol discussion entirely and use the data simply to demonstrate a negative result consistent with our model.

*5) The somewhat grander claims regarding the implied role these LXG proteins play in polymicrobial communities is perhaps a bit over-speculative. In general, it seem that there is a disconnect between the data the authors present and their emphasis in the Abstract (and throughout the manuscript) on human gut metagenomes and polymicrobial dynamics. Throughout the manuscript, the authors emphasize the abundance/prevalence of LXG proteins in human microbiomes, but those data are neither very convincing nor novel. Thus, the authors have overstated the implications of the inter-species antagonism mediated by Esx secretion, as they do not present data illustrating the functional outcome of these interactions in an actual polymicrobial community.*

At the suggestion of the reviewers, we have toned down and limited references in the main body of the revised manuscript to the potential role of LXG toxins in human-associated polymicrobial communities. We have also edited the Abstract with the intent to minimize this potential implication of the prevalence of LXG toxins amongst human gut microbiome samples.

We feel that we must respectfully voice our disagreement with the reviewer comment pertaining to our metagenomic analyses that “those data are neither very convincing nor novel.” To our knowledge, LXG toxins have not previously been identified in the human gut microbiome (or another polymicrobial community) and we employed rigorous, accepted methods of showing that they are present in said communities in our study.

*6) The authors conclude that LXG toxins are specific to Gram + inter-species antagonism, but have only tested a single Gram negative species (E. coli). Testing a broader range of Gram negatives would be an easy way to bolster this claim.*

First, to clarify, we respectfully wish to correct the record by pointing out that we did not “conclude that LXG toxins are specific to Gram + inter-species antagonism.” The single statement we made addressing these data in our original submission, stated “These results demonstrate that the Esx pathway can act between species and suggest that its target range may be limited to Gram-positive bacteria.” We specifically chose the words “suggest” and “may” in this statement because we agree that stronger terms are difficult (or even impossible) to support.

At the suggestion of the reviewers, the revised manuscript now includes the results of experiments testing the capacity of the Esx secretion pathway to target additional Gram-negative bacteria. We report that under conditions similar to those used in the corresponding Gram-positive target cell experiments, the Esx system of *S. intermedius* did not detectably alter the outcome of co-cultivations with *P. aeruginosa, B. thailandensis* and *Bacteroides fragilis* (new Figure 3). *B. thailandensis* and *B. fragilis*, a β-proteobacterium and a member of the phylum Bacteroidetes, respectively, significantly extend the phylogenetic range of Gram-negative organisms tested. While we now report more Gram-negative bacteria tested, we recognize that this does not bring us significantly closer to reaching the conclusion that Gram-negative bacteria are not or cannot be targeted by the Esx pathway. We thus have not modified our statement that “These results demonstrate that the Esx pathway can act between species and suggest that its target range may be limited to Gram-positive bacteria”.

*7) The notion that TelB is a NAD+ degrading toxin is less than compelling. Two important controls are missing in Figure 2 – mutation of the conserved residues that comprise the active site as predicted in Figure 2—figure supplement 2, and rescue of the toxic effect by co-expressing TipB. Complementation of the Si telB mutant using wild-type and mutant forms of TelB would also be important in this regard as well. Finally, the assay in Figure 2 is somewhat unconvincing. An* in vitro *NADase activity assay with purified TelB would be much more convincing and quantitative.*

In our original manuscript, we used two independent biochemical methods to demonstrate that TelB is an NAD^+^-degrading toxin. The first makes use of the fact that if NAD^+^ is not consumed by such a toxin, a product of its hydrolysis by base is fluorescent (Johnson et al. J. Biol. Chem. 1970). This assay showed that wild-type *S. intermedius* B196 possesses supernatant NAD^+^-degrading activity that is entirely dependent on TelB and EssC (Figure 2). We also showed that another strain of *S. intermedius* that does not encode a TelB ortholog does not possess this activity in its supernatant (Figure 2). The second biochemical assay we presented makes use of a very well established, specific, and sensitive cell-based assay for measuring NAD^+^ pools that is commercially available. We used this assay to show that induction of TelB and an established NADase, Tse6, lead to a dramatic drop in cellular [NAD+], whereas induction of an interbacterial toxin that does not possess this activity, Tse2, does not reduce cellular NAD^+^ levels (Figure 2). In addition to these specific biochemical assays, we stated that “One such toxin domain shares homology and predicted catalytic residues with *M. tuberculosis* TNT, an NAD+-degrading enzyme” and presented a molecular model of TelB based on the TNT structure that showed conserved catalytic residues. Indeed, we noted in the methods section of the original submission that, due to its potent toxicity, we were unable to clone TelB_tox_, and “to circumvent this, SOE PCR was used to assemble a variant (H661A) that was tolerated under non-induced conditions. Based on the similarity of TelB_tox_ to *M. tuberculosis* TNT toxin, this mutation likely reduces the binding affinity of TelB to NAD^+^ (Sun et al., 2015).”

For technical reasons relating to its stability, we are unable to purify TelB. These enzymes are well characterized and we do not believe that a detailed quantification of its activity is necessary to support the conclusions of this manuscript. However, at the request of the reviewer, we now include data demonstrating that the NAD^+^-degrading activity of TelB is abrogated by point mutations in the active site of the enzyme and by co-expression with TipB (new Figure 2).

*8) Given that the main emphasis in the paper is on TelC, it is disappointing that the authors do not demonstrate inter-species antagonism dependent on TelC. Perhaps we missed something, but it appears that the data in Figure 2 are all experiments done with S.i. strains. In Figure 2, they show that a strain WT for TelB elicits the crucial inter-species antagonism, but it is not clear whether this strain is also making TelC?*

We agree with the reviewers and the revised manuscript now includes the finding that TelC contributes significantly to the fitness of *S. intermedius* during interspecies competition (new Figure 4—figure supplement 3). As a point of clarification, the “wild-type” strain used in Figure 2 of our original submission was indeed wild-type and thus possesses and presumably utilizes its full complement of LXG effectors.

*9) The idea that all three LXG proteins are secreted from Si via Esx secretion is not well supported. Table 1 is confusing – the "locus tag" is not intuitive, making it difficult to understand which proteins are represented in the MS results. Importantly, it is critical to show that secreted proteins that are not associated with Esx are present in similar levels in wild-type and essC mutant supernatants.*

We regret the lack of clarity in Table 1 of our original submission. At the suggestion of the reviewers, we have added a column to Table 1 that provides the common names of the proteins we detected. Importantly, SIR_0169, SIR_0179, and SIR_1489 are now identified as TelA, TelB, and TelC, respectively (note – this information was provided in Figure 2 of the original submission). There additionally may have been some confusion regarding the footnote that defined the values we provided for each entry in Table 1. These values represent the average spectral counts – a semi-quantitative correlate of protein abundance from mass spectrometry data (Liu et al.,Anal. Chem. 2004) – for each protein from triplicate biological replicates. The extracellular proteome of *S. intermedius* has not previously been defined; however, the wild-type:∆*essC* ratio of these spectral count values clearly indicates that known Esx-associated proteins and the Tel proteins are underrepresented in the extracellular proteome of ∆*essC* relative to 29 of the 30 other proteins we identified.

*10) Figure 4 is compelling but underdeveloped. The exclusive use of bacterial two-hybrid seems insufficient to show convincingly that the LSG and WXG proteins interact specifically, and thus any conclusions at this point seems speculative.*

We are unsure if there was disagreement amongst the reviewers as to whether the data in Figure 4 of our original submission was compelling or underdeveloped. Nevertheless, we agree that the exclusive use of the bacterial two-hybrid to demonstrate an interaction would be problematic and for this reason we included in vivo functional data in the original submission (Figure 4) that supports the interaction between LXG proteins and cognate WXG-like proteins. In short, we showed that a deletion of *wxgC* abrogates TelC secretion while leaving TelB secretion intact. To further support this, we added to the revised manuscript an experiment demonstrating that in a heterologous host, TelC immunoprecipitates WxgC and not WxgB – additional evidence of their direct interaction (new Figure 5). In summary, we now provide genetic, biochemical, and in vivo functional data supporting the specific interaction of Lxg proteins with cognate Wxg proteins.

*11) In Figure 2, why does the CI decrease to well below 1 when the filter is present?*

This is an excellent question that we have clarified in the revised manuscript (see revised legend to Figure 3). In the experiments represented, recipient cell populations were placed below the filter, whereas the donor cells were above the filter. As the cells proliferate, those below the filter have an Esx-independent growth advantage by virtue of their immediate proximity to the growth substrate. It is worth noting that we conducted this experiment in both configurations and in neither case were non-contacting cells inhibited in an Esx-dependent manner.

*12) Roughly how much TelC is secreted in an exponential or overnight culture? It is not clear how to interpret Figure 2—figure supplement 3 where bacteria are incubated with purified "active toxin" without an estimate of the amount of toxin produced endogenously. Also, the authors don't show data to indicate that their purified recombinant TelC protein is active.*

As we noted above, TelC (and the other Tel proteins) are of lower than average abundance (as determined by spectral counts) among the 35 proteins identified in our secreted proteome analysis (Table 1). We are not aware of absolute measurements of secreted protein concentration during bacterial culture, but we estimate based on our own experience studying secreted proteins that those we can detect are present at 5-100 ng/mL. We choose a concentration of 0.1 mg/mL of TelC in order to ensure that the protein was present at levels exceeding those obtainable physiologically. To probe this directly and address the question raised, we present here a comparison of the relative levels of endogenous TelC to the levels of purified TelC we added exogenously to *S. intermedius* cells. As expected, endogenous TelC is below the detection limit of this assay, whereas 0.1 mg/mL purified TelC is readily detected (Figure 9).

Author response image 4.TelC levels in cell free supernatants (CFS) of S. intermedius do not approach 0.1 mg/mL.SDS-PAGE analysis of the supernatant fraction of *S. intermedius* cells in mid-log or stationary growth. To examine the relative levels of endogenous TelC versus the exogenous purified active TelC we applied, supernatants were doped with 0.1 mg/mL TelC (CFS + TelC) and the protein was visualized alongside CFS alone.**DOI:**
http://dx.doi.org/10.7554/eLife.26938.025

With regard to the question pertaining to the activity of purified recombinant TelC, we first wish to clarify that in our original submission, all biochemical assays demonstrating the capacity of this protein to act on undecaprenylphosphate derivatives were performed using purified recombinant protein. Though it is not specified here, the reviewers may be referring to the fact that our initial biochemical assays were performed with the toxin domain of TelC (TelC_tox_), in contrast to the full-length protein used in our cellular inhibition assays. We use full-length TelC in lieu of TelC_tox_ for growth inhibition assays to allow potential cell entry mechanisms that could utilize the N-terminal domains of the protein. At the suggestion of the reviewers, we have performed biochemical assays with purified full-length TelC. New Figure 4—figure supplement 3 shows that like the toxin domain alone, the full-length protein possesses the capacity to degrade lipid II.

*13) In Figure 1, the authors highlight another Streptococcus species (parasanguinus) that has an LXG protein which they annotate as a Lipid II phosphatase. That raises the question of how novel their finding about TelC in Si is? This deserves clarification.*

The *S. parasanguinus* protein schematized in Figure 1 is a predicted ortholog of TelC and was assigned a predicted activity based solely on this relatedness. Were an LXG protein possessing lipid II phosphatase activity known, we certainly would have highlighted and cited such a highly relevant result. To prevent this potential confusion, we have modified the Figure 1 legend to read: “lipid II phosphatase based on orthology to TelC (defined biochemically herein)” in place of “TelC-like lipid II phosphatase” found in the original manuscript.

[Editors' note: further revisions were requested prior to acceptance, as described below.]

*Reviewer #2:*

*The manuscript has improved significantly because of the re-write and the inclusion of new data. I appreciate the authors' detailed responses to the reviews and their efforts to address the issues. Overall, I think they addressed the main concerns satisfactorily. I would only like to request a few modifications to the text.*

*The main comment I have is about the cellular target of TelC. The authors show that TelC: a) needs to be exported to be toxic in both E. coli and Staph, and b) degrades both lipid II and undecaprenyl-PP (C55-PP)* in vitro*. They conclude that, in cells, TelC targets lipid II that is located in the outer leaflet of the cytoplasmic membrane. This could lead readers to mistakenly think that lipid II is only present at that location and that C55-PP is not. In reality, C55-PP is also present in the outer leaflet of the cytoplasmic membrane, and both lipid II and C55-PP are present in both leaflets of the cytoplasmic membrane. Therefore, I suggest that the authors state that their* in vivo *data cannot distinguish whether TelC targets lipid II and/or undecaprenyl-PP (C55-PP) and that it is unknown why toxicity is limited to an extracytoplasmic location given that these essential peptidoglycan intermediates transit through both leaflets of the cytoplasmic membrane. [A simple explanation for their data could be that the intermediate(s) is not accessible in the cytoplasmic side of the membrane because proteins that normally handle it during peptidoglycan synthesis protect it from TelC or the intermediate has a very short residence time at that location].*

At the suggestion of the reviewer we have added text stating that C55-PP and lipid II are present both inside and outside of the cell, that thus that it is unknown why cytoplasmic TelC is non-toxic. In this text, we acknowledge the substrate accessibility explanation mentioned by the reviewer and, in addition, we note that the abundance of free calcium ions outside of the bacterial cell may be important for TelC folding.